# *Jump Your Steps*: Optimizing Sampling Schedule of Discrete Diffusion Models

**Yonghyun Park**\* **& Chieh-Hsin Lai**
Sony AI
Tokyo, Japan
`enkeejunior1@snu.ac.kr`

**Satoshi Hayakawa**
Sony Group Corporation
Tokyo, Japan

**Yuhta Takida**
Sony AI
Tokyo, Japan

**Yuki Mitsufuji**
Sony AI, Sony Group Corporation
New York, USA

## Abstract

Diffusion models have seen notable success in continuous domains, leading to the development of discrete diffusion models (DDMs) for discrete variables. Despite recent advances, DDMs face the challenge of slow sampling speeds. While parallel sampling methods like $\tau$-leaping accelerate this process, they introduce *Compounding Decoding Error* (CDE), where discrepancies arise between the true distribution and the approximation from parallel token generation, leading to degraded sample quality. In this work, we present *Jump Your Steps* (JYS), a novel approach that optimizes the allocation of discrete sampling timesteps by minimizing CDE without extra computational cost. More precisely, we derive a practical upper bound on CDE and propose an efficient algorithm for searching for the optimal sampling schedule. Extensive experiments across image, piano note, and text generation show that JYS significantly improves sampling quality, establishing it as a versatile framework for enhancing DDM performance for fast sampling. The code is available at `https://github.com/sony/jys`.

## 1 Introduction

Diffusion models (Sohl-Dickstein et al., 2015; Song et al., 2021b; Ho et al., 2020; Song et al., 2021a; Karras et al., 2022) have achieved remarkable success in generation tasks within the continuous domain. However, certain modalities, such as text and piano note, inherently possess discrete features. Recently, discrete diffusion models (DDMs) (Austin et al., 2021; Campbell et al., 2022; 2024; Gat et al., 2024) have demonstrated performance comparable to state-of-the-art methods in various areas, including text (Lou et al., 2024; Shi et al., 2024) and image (Chang et al., 2022; Gu et al., 2022) generation. Nevertheless, like their continuous counterparts, DDMs encounter a significant bottleneck in sampling speed due to their progressive refinement process.

In contrast to continuous-domain diffusion models, where sampling dynamics are driven by sample-wise differential equations (Song et al., 2021b), allowing for the direct application of well-established numerical methods to accelerate generation, enhancing speed in DDMs poses a significant challenge. To address this, researchers have proposed fast and efficient samplers, including notable methods such as the $\tau$-leaping (Campbell et al., 2022; Lezama et al., 2022; Sun et al., 2023) and $k$-Gillespie algorithms (Zhao et al., 2024), which facilitate parallel sampling of multiple tokens in a single step. However, this parallel but independent sampling introduces *Compounding Decoding Error* (CDE) (Lezama et al., 2022), which arises from a mismatch between the training and inference distributions of intermediate latents during parallel sampling. Specifically, while each token is generated according to its marginal distribution, the joint distribution deviates from the learned distribution. To mitigate this issue, the predictor-corrector (PC) sampler (Campbell et al., 2022) has been proposed. This sampler slightly perturbs the generated data to correct incorrectly generated tokens. However, these methods have limitations, including impracticality under low computational

---

\*Work done during an internship at SONY AI.

budgets (Campbell et al., 2022), the need for an additional corrector (Lezama et al., 2022), or reliance on specialized architectures and loss functions (Zhao et al., 2024).

Figure 1: *(Top)* Comparison of sampling trajectories: ground truth vs. parallel sampling using a uniform schedule and the Jump Your Steps (JYS) schedule. *(Bottom)* Uniform schedule exhibits compounding decoding errors during parallel sampling, while JYS reduces them by using fewer steps in deterministic phases and reallocating skipped steps to other timesteps.

To reduce CDE and enable fast sampling in DDM fundamentally, we first introduce a rigorous quantity to measure CDE (see Figure 1 Top, and Section 3.1) and propose a novel approach called *Jump Your Steps* (JYS), which optimizes the allocation of discrete sampling timesteps $\{T \rightarrow t_1 \rightarrow \ldots \rightarrow t_{N-1} \rightarrow 0\}$[1] under a fixed total sampling budget $N$ to minimize CDE. Our core idea is to derive efficiently computable bounds for CDE (see Section 3.2 and Section 3.3) and strategically select sampling timesteps by solving minimization problems to reduce these bounds (Section 3.4 and Section 3.5), theoretically ensuring a decrease in the gap between the ground truth distribution and the approximated distribution through parallel sampling (see Figure 2).

Unlike previous methods such as the PC sampler, our approach requires no additional computational resources or modifications to the model architecture or loss function. We empirically validate the effectiveness of our sampling schedule across various datasets, including synthetic sequential data, CIFAR-10 (image), Lakh Pianoroll (piano note), and text modeling. Our approach accelerates DDM sampling across models using different forward corruption transition kernels, such as uniform, Gaussian, and absorbing transition matrices. Our comprehensive experiments cover both unconditional and conditional generation tasks, consistently showing that optimizing the sampling schedule significantly enhances sampling quality. These results indicate that our method serves as a general framework for speeding up discrete diffusion model sampling.

## 2 BACKGROUND

### 2.1 CONTINUOUS TIME FRAMEWORK FOR DISCRETE DIFFUSION MODELS.

DDMs define the generative process as the reverse of the data-corrupting forward process, expressed as a Continuous Time Markov Chain (CTMC) on a finite state space $\mathcal{S}$ (Campbell et al., 2022). For the data-corrupting process $(X_t)_{t \in [0,T]}$, the density evolution is described as:

$$q_{t+dt|t}(y \mid x) = \delta_{xy} + R_t(x, y)dt + o(dt) \tag{1}$$

Here, $\delta_{xy}$ is the Dirac delta function, $R_t \in \mathbb{R}^{S \times S}$ is the transition rate matrix of the forward CTMC, with $S = |\mathcal{S}|$, and $dt > 0$. Rate matrices ensure the marginal distribution $q_t(x_t) =$

---

[1]In this work, $s \rightarrow t$ denotes sampling across timesteps from $s$ to $t$.

$\int q_t(x_t|x_0)q_0(x_0)dx_0$, where $q_0 = p_{\text{data}}$ and $q_T \approx \pi$, the stationary distribution of the forward CTMC. Various transition matrices have been proposed, allowing $\pi$ to follow a uniform or Gaussian distribution, or converting samples into masked tokens.

For generation, we reverse the forward process, moving from the marginal $q_T$ back to $p_{\text{data}}$. This time-reversal CTMC is also a CTMC (Campbell et al., 2022; 2024):

$$q_{t-dt|t}(y \mid x) = \delta_{xy} + \tilde{R}(x, y)dt + o(dt), \tag{2}$$

where the backward transition rate $\tilde{R}$ is defined as:

$$\tilde{R}(x, y) = R(y, x) \underbrace{\frac{q_t(y)}{q_t(x)}}_{\text{Score Parametrization}} = R(y, x) \sum_{x_0} \frac{q_t(y \mid x_0)}{q_t(x \mid x_0)} \underbrace{q_{0|t}(x_0 \mid x)}_{\text{Denoising Parametrization}}$$

The literature primarily falls into two parameterizations: Denoising parameterization (Campbell et al., 2024; Austin et al., 2021; Campbell et al., 2022) approximates a parameterized denoising model as $p^\theta_{0|t}(x_0|x) \approx q_{0|t}(x_0|x)$. Conversely, score parameterization (Lou et al., 2024; Meng et al., 2022) estimates the ratio of the data distribution as $s^\theta_t(y|x) = q_t(y)/q_t(x)$.

## 2.2 SAMPLING FROM THE BACKWARD CTMC

**Gillespie's Algorithm**  was proposed as a simulation algorithm for a given CTMC (Gillespie, 2007). Gillespie's algorithm simulates the CTMC by calculating the rate matrix at each state transition. If the rate matrix of the CTMC depends only on the state, Gillespie's algorithm serves as an exact simulation method. However, since it allows for only one token transition each time the rate matrix is calculated, it is computationally inefficient.

$k$**-Gillespie's Algorithm**  Instead of updating only one token for each rate matrix calculation, the $k$-Gillespie's algorithm (Zhao et al., 2024) updates $k$ tokens in parallel. This reduces the computation by a factor of $1/k$ compared to the original Gillespie algorithm.

$\tau$**-Leaping**  On the other hand, Campbell et al. (2022) proposes sampling through $\tau$-leaping. Unlike the $k$-Gillespie algorithm, which update $k$ tokens in parallel, $\tau$-leaping simultaneously updates all tokens according to the given fixed rate matrix within the specified time interval $[t, t + \tau)$. Recently, Tweedie $\tau$-leaping, which considers changes in the rate matrix according to the noise schedule, has been proposed (Sun et al., 2023; Lou et al., 2024).

## 3 OPTIMIZING THE SAMPLING SCHEDULE OF DISCRETE DIFFUSION MODELS

In this section, we aim to optimize sampling schedule $\{T \to t_1 \to t_2 \to \ldots \to t_{N-1} \to 0\}$ to minimize the CDE introduced by parallel sampling. First, we define and analyze the CDE, examining its relationship to both the sampling schedule (Section 3.1) and sampling quality (Section 3.2). In Section 3.3, we derive an upper bound on the CDE, which serves as the objective for the sampling schedule optimization. Finally, we introduce heuristic algorithms to make the optimization tractable including hierarchical breakdown strategy (Section 3.4) and computational techniques (Section 3.5). Figure 2 summarizes the relationships between the theoretical analyses and algorithms discussed in this section.

Although this section focuses on samplers based on $\tau$-leaping, all methods are also applicable to the $k$-Gillespie algorithm. For extensions to $k$-Gillespie, please refer to Algorithm 1.

**Notations**  To begin, we introduce some essential mathematical notation. $X$ : a random variable, $\mathbf{x}$ : its observation, $\mathbb{P}, \mathbb{Q}$ : distributions, $\{T \to t_1 \to \ldots \to t_{N-1} \to 0\}$ : sampling schedule, and $\mathbb{Q}^{a \to b \to \cdots \to c}$ : the distribution generated by the sampling schedule $\{a \to b \to \cdots \to c\}$. For clarity, when working with backward CTMCs, we slightly abuse notation and express intervals as $[s, t] \triangleq \{u \mid s \geq u \geq t\}$; the same applies to open and half-open intervals.

$$\underset{\text{Theorem 3.1}}{\mathcal{D}_{\mathrm{KL}}(\mathbb{P}_0\|\mathbb{Q}_0^{t_0 \rightarrow t_1 \rightarrow \cdots \rightarrow 0})} \quad \leq \quad \sum_{i=0}^{N-1}\mathcal{D}_{\mathrm{KL}}(\mathbb{P}_{t_{i+1}}\|\mathbb{Q}_{t_{i+1}}^{t_i \rightarrow t_{i+1}}) \quad \leq \quad \underset{\text{Eq. (10)}}{\mathcal{D}_{\mathrm{KL}}(\mathbb{P}_{\mathrm{paths}}\|\mathbb{Q}_{\mathrm{paths}}^{t_0 \rightarrow t_1 \rightarrow \cdots \rightarrow 0})}$$

Eqs. (3, 5)        Theorem 3.2

$$\sum_{i=0}^{N-1}\mathcal{E}_{\mathrm{CDE}}(t_i \rightarrow t_{i+1}) \qquad \mathrm{KLUB}_{\mathbb{P}_T}(\mathbb{P}_0\|\mathbb{Q}_0^{t_0 \rightarrow t_1 \rightarrow \cdots \rightarrow 0})$$

Algorithm 1, 2.

$$\{T \rightarrow t_1 \rightarrow \ldots \rightarrow t_{N-1} \rightarrow 0\}$$

Figure 2: An illustration of the relationship between the KL divergence of the distribution, the compounding error $\mathcal{E}_{\mathrm{CDE}}$ (defined in Section 3.1), and KLUB (defined in Section 3.3). The sampling schedule $\{T \rightarrow t_1 \rightarrow \ldots \rightarrow t_{N-1} \rightarrow 0\}$ is optimized to minimize KLUB using the efficient algorithms detailed in Section 3.4, and 3.5.

## 3.1 TIME-DEPENDENT NATURE OF COMPOUNDING DECODING ERRORS

We introduce a measure for the CDE, $\mathcal{E}_{\mathrm{CDE}}$, which quantifies the discrepancy between the true joint distribution and the distribution from parallel token generation. For illustration, we consider a discrete process $X_t = (X_t^1, X_t^2)$ with sequence length 2, consisting of tokens $X_t^1$ and $X_t^2$. The general case is provided in Appendix B.1.

We propose measuring the CDE for a single parallel sampling step $\{s \rightarrow t\}$ start from $\mathbf{x}_s$ by using the KL divergence between the joint distribution $P_{X_t^1, X_t^2|\mathbf{x}_s}$ and the product of marginal distributions $P_{X_t^1|\mathbf{x}_s} \otimes P_{X_t^2|\mathbf{x}_s}$:

$$\mathcal{E}_{\mathrm{CDE}}(s \rightarrow t|\mathbf{x}_s) \triangleq \mathcal{D}_{\mathrm{KL}}(\underbrace{P_{X_t^1, X_t^2|\mathbf{x}_s}}_{\text{True distribution}} \| \underbrace{P_{X_t^1|\mathbf{x}_s} \otimes P_{X_t^2|\mathbf{x}_s}}_{\text{Approx. distribution from parallel sampling}}). \tag{3}$$

We note that the defined CDE is equivalent to the conditional mutual information $\mathcal{I}(X_t^1; X_t^2|\mathbf{x}_s)$ of tokens $X_t^1, X_t^2$:

$$\mathcal{E}_{\mathrm{CDE}}(s \rightarrow t|\mathbf{x}_s) = \mathcal{I}(X_t^1; X_t^2|\mathbf{x}_s). \tag{4}$$

This expression links the compounding error to the mutual information between tokens; lower mutual information reduces parallel sampling errors. For example, as shown in Figure 1 *(Bottom)*, as generation progresses, the uncertainty of each token decreases over time due to the tokens already generated, reducing mutual information and preventing CDE. In general, CDE depends on the timesteps, and its behavior varies with the data distribution, corruption kernel (see Fig. 9 for illustration), and DDM sampling methods. Motivated by this observation, we hypothesize that we can reduce the CDEs during generation process by optimizing the sampling schedule.

## 3.2 RELATION BETWEEN COMPOUNDING DECODING ERRORS AND GENERATION QUALITY

While the Eq. (3) allows us to estimate the CDE starting from a specific state $\mathbf{x}_s$, in practice, we are interested in the average compounding error over all possible starting states at time $s$. To assess the overall impact of the CDE when transitioning over the timesteps $s \rightarrow t$, we consider the expected value of $\mathcal{E}_{\mathrm{CDE}}(s \rightarrow t|\mathbf{x}_s)$ with respect to $\mathbf{x}_s \sim \mathbb{P}_s$. This leads us to consider:

$$\mathcal{E}_{\mathrm{CDE}}(s \rightarrow t) \triangleq \mathbb{E}_{\mathbf{x}_s}\left[\mathcal{E}_{\mathrm{CDE}}(s \rightarrow t|\mathbf{x}_s)\right]. \tag{5}$$

Consider a sampling schedule $\{T = t_0 \rightarrow t_1 \rightarrow \ldots \rightarrow t_{N-1} \rightarrow 0 = t_N\}$, which will be specified later. Our goal is to minimize the cumulative CDE that arises from each parallel sampling step within the given schedule. If we ignore the accumulated error from the previous steps that affects the consecutive steps, our objective is as follows (Appendix A.2):

$$\min_{t_1, t_2, \ldots, t_{N-1}} \sum_{i=0}^{N-1} \mathcal{E}_{\mathrm{CDE}}(t_i \rightarrow t_{i+1}). \tag{6}$$

Interestingly, we find in the following theorem that cumulative CDEs over the sampling schedule can upper bound the KL divergence between the true distribution at time $t = 0$, denoted $\mathbb{P}_0$, and the distribution $\mathbb{Q}_0^{T \to t_1 \to \cdots \to 0}$ obtained from parallel sampling along the sampling schedule (see Appendix B.1 for proof):

**Theorem 3.1.** *We have the following bound on the KL divergence between $\mathbb{P}_0$ and $\mathbb{Q}_0^{T \to t_1 \to \cdots \to 0}$ in terms of cumulative CDEs:*

$$\mathcal{D}_{\mathrm{KL}}(\mathbb{P}_0 \| \mathbb{Q}_0^{T \to t_1 \to \cdots \to 0}) \leq \sum_{i=0}^{N-1} \mathcal{E}_{\mathrm{CDE}}(t_i \to t_{i+1}). \tag{7}$$

This theorem suggests that effectively allocating the time schedule to minimize the CDEs can implicitly reduce the discrepancy between the true distribution and the approximate distribution obtained from parallel sampling. Motivated by this, in the next section, we derive a tractable upper bound for $\sum_{i=0}^{N-1} \mathcal{E}_{\mathrm{CDE}}(t_i \to t_{i+1})$ that depends on the sampling schedule $\{T \to t_1 \to \ldots \to t_{N-1} \to 0\}$ to facilitate its optimization.

## 3.3 ESTIMATING THE COMPOUNDING DECODING ERROR USING GIRSANOV'S THEOREM

As shown in Eq. (3), computing $\mathcal{E}_{\mathrm{CDE}}(s \to t | \mathbf{x}_s)$ involves determining the KL divergence between the true distribution and the approximated distribution from DDM's parallel sampler, which is often intractable. To address this, we treat the ground truth reverse process and the sampling process from DDM's parallel samplers (introduced in Section 2.2) as two CTMCs, starting from the same initial distribution. By applying **Girsanov's theorem** (Ding & Ning, 2021; Chen et al., 2023), we derive a tractable formula to compare the KL divergence between the distributions of these stochastic processes at any time interval $[s, t]$. We summarize this as the following general theorem applicable to any two backward CTMCs with $R_t^1$ and $R_t^2$ as their respective transition rate matrices:

$$\begin{cases} \text{CTMC 1}: & q_{u-du|u}^1(y \mid x) = \delta_{xy} + R_t^1(x, y)du + o(du), \\ \text{CTMC 2}: & q_{u-du|u}^2(y \mid x) = \delta_{xy} + R_t^2(x, y)du + o(du). \end{cases}$$

We defer its proof to Appendix B.2.

**Theorem 3.2.** (KL-Divergence Upper Bound, KLUB) *Consider an interval $[s, t]$ ($s > t$). If both CTMCs start from the same initial distribution, $\pi_s = \mathbb{P}_s = \mathbb{Q}_s$, then we have:*

$$\mathcal{D}_{\mathrm{KL}}(\mathbb{P}_t \| \mathbb{Q}_t) \leq \mathcal{D}_{\mathrm{KL}}(\mathbb{P}_{\mathrm{paths}} \| \mathbb{Q}_{\mathrm{paths}}) = \underbrace{\mathbb{E}_{\mathbb{P}_{\mathrm{paths}}} \left[ \sum_{i \neq j} \sum_{t < u \leq s} H_u^{ij} \log \frac{R_u^1(i, j)}{R_u^2(i, j)} \right]}_{\triangleq \mathrm{KLUB}_{\pi_s}(\mathbb{P}_t \| \mathbb{Q}_t)}. \tag{8}$$

*Here, $\mathbb{P}_t$ and $\mathbb{Q}_t$ are the probability distributions at time $t$ resulting from CTMC 1 and CTMC 2, respectively. $\mathbb{P}_{\mathrm{paths}}$ and $\mathbb{Q}_{\mathrm{paths}}$ denote the distributions over their path spaces $(X_u)_{u \in [t, s]}$, generated by CTMC 1 and CTMC 2, respectively. The indicator function $H_u^{ij}$ is defined as $H_u^{ij} = 1$ if a transition from state $i$ to $j$ occurs at time $u$, and $H_u^{ij} = 0$ otherwise.*

We denote the rightmost term in Eq. (8) as the *Kullback-Leibler Divergence Upper Bound (KLUB)*, which quantifies the mismatch between distributions generated by different CTMCs based on their rate matrices. Consider CTMC 1 as the ground truth reverse CTMC and CTMC 2 as the reverse CTMC obtained via parallel sampling by substituting the forward transition kernel with the corresponding reverse-in-time kernel. Thus, $\mathcal{E}_{\mathrm{CDE}}(s \to t | \mathbf{x}_s)$ can be expressed as the KL divergence between the output distributions of the two CTMCs at time $t$, starting from the initial point $\mathbf{x}_s$. This leads to the upper bound (Proof in Appendix B.3):

$$\mathcal{E}_{\mathrm{CDE}}(s \to t) \leq \mathrm{KLUB}_{\mathbb{P}_s}(\mathbb{P}_t \| \mathbb{Q}_t^{s \to t}), \tag{9}$$

where $\mathbb{Q}_t^{s \to t}$ is given by the process discretized at time $s$ and $t$. This expectation considers the distribution of states at time $s$ and offers a more comprehensive measure of the CDE over the interval $[s, t]$. Moreover, from the additivity of KLUB, we can bound the sum of CDEs as

$$\mathcal{E}_{\mathrm{CDE}}(s \to t) + \mathcal{E}_{\mathrm{CDE}}(t \to u) \leq \mathrm{KLUB}_{\mathbb{P}_s}(\mathbb{P}_t \| \mathbb{Q}_t^{s \to t}) + \mathrm{KLUB}_{\mathbb{P}_t}(\mathbb{P}_u \| \mathbb{Q}_u^{t \to u}) = \mathrm{KLUB}_{\mathbb{P}_s}(\mathbb{P}_u \| \mathbb{Q}_u^{s \to t \to u}), \tag{10}$$

which shows that the KLUB can be useful for comparing the quality of discretization of the interval $[s, u]$ with different break point $t$. We can easily extend this result for the sum of CDEs over the entire sampling schedule $\{T = t_0 \to t_1 \to \ldots \to t_{N-1} \to 0 = t_N\}$.

$$\mathcal{D}_{\mathrm{KL}}(\mathbb{P}_0 \| \mathbb{Q}_0^{T \to t_1 \to \cdots \to 0}) \leq \sum_{i=0}^{N-1} \mathcal{E}_{\mathrm{CDE}}(t_i \to t_{i+1}) \leq \mathrm{KLUB}_{\mathbb{P}_T}(\mathbb{P}_0 \| \mathbb{Q}_0^{T \to t_1 \to \cdots \to 0}), \tag{11}$$

where inequality on the left-hand side comes from Theorem 3.1.

To summarize, optimizing the sampling schedule involves finding a set of timesteps $\{T \to t_1 \to \ldots \to t_{N-1} \to 0\}$ that minimizes the KLUB on the right-hand side. This approach approximately reduces the cumulative CDE (middle) and provides an upper bound on the KL divergence between the true distribution and the sampled distribution for the given schedule (left-hand side).

### 3.4 FEASIBLE COMPUTATION WITH HIERARCHICAL BREAKDOWN STRATEGY

Using the derived KLUB, we can formulate the timestep search as a minimization problem over KLUB. Here, we employ a hierarchical breakdown strategy, dividing a coarser sampling schedule into a finer one, as shown in Figure 3.4. Suppose our sampling schedule is given by $\{T \to t \to 0\}$. Let $\mathbb{Q}_0^{T \to t \to 0}$ represent the distribution generated by this schedule. Our goal is to find the optimal $t$ that minimizes cumulative CDE, i.e., $\mathcal{E}_{\mathrm{CDE}}(T \to t) + \mathcal{E}_{\mathrm{CDE}}(t \to 0)$. This is approximately achievable by minimizing its KLUB upper bound:

$$t_1 = \underset{t \in (T, 0)}{\arg \min} \, \mathrm{KLUB}(\mathbb{P}_0 \| \mathbb{Q}_0^{T \to t \to 0}) \tag{12}$$

With the initial refined interval $\{T \to t_t \to 0\}$, we seek optimal timesteps $t_2 \in (T, t_1)$ and $t_3 \in (t_1, 0)$ by solving the following minimization problems:

$$t_2 \in \underset{t \in (T, t_1)}{\arg \min} \, \mathrm{KLUB}(\mathbb{P}_{t_1} \| \mathbb{Q}_{t_1}^{T \to t \to t_1}) \quad \text{and} \quad t_3 \in \underset{t \in (t_1, 0)}{\arg \min} \, \mathrm{KLUB}(\mathbb{P}_0 \| \mathbb{Q}_0^{t_1 \to t \to 0}).$$

The first minimization problem targets matching $\mathcal{D}_{\mathrm{KL}}(\mathbb{P}_{t_1} \| \mathbb{Q}_{t_1})$, while the second focuses on matching $\mathcal{D}_{\mathrm{KL}}(\mathbb{P}_0 \| \mathbb{Q}_0)$. This results in a further refined sampling schedule $\{T \to t_2 \to t_1 \to t_3 \to 0\}$. By iterating this process, we continue splitting each interval into smaller ones, optimizing breakpoints using the KLUB criterion. After $K$ iterations, this hierarchical strategy yields a sampling schedule with $2^K$ NFEs (Number of Function Evaluations), optimizing the schedule as the number of steps increases.

Figure 3: We optimize the sampling schedule by refining it from coarse intervals to finer intervals, using a hierarchical breakdown strategy.

### 3.5 FEASIBLE COMPUTATION FOR KLUB ESTIMATION

Directly estimating $\mathrm{KLUB}_{\mathbb{P}_T}(\mathbb{P}_0 \| \mathbb{Q}_0^{T \to t \to 0})$ using Eq. (8) is impractical due to (1) the high cost of sampling from $\mathbb{P}_{\mathrm{paths}}$, and (2) the need for extensive Monte Carlo sampling for reliable estimates. To address this issue, we propose two techniques that simplify the estimation process.

**Technique 1: Maximizing KL Divergence from a Coarser Approximation** Instead of minimizing the mismatch between the ground truth distribution $\mathbb{P}_{\mathrm{paths}}$ and $\mathbb{Q}_{\mathrm{paths}}^{T \to t \to 0}$, we choose to maximize the discrepancy between $\mathbb{Q}_{\mathrm{paths}}^{T \to t \to 0}$ and a simpler, coarser approximation $\mathbb{Q}_{\mathrm{paths}}^{T \to 0}$. The key insight is that maximizing this divergence, we can find the optimal sampling time $t$, which helps reduce the compounding error relative to the true distribution. This relationship can be approximated as (Appendix B.4):

$$\mathcal{D}_{\mathrm{KL}}(\mathbb{Q}_{\mathrm{paths}}^{T \to t \to 0} \| \mathbb{Q}_{\mathrm{paths}}^{T \to 0}) \approx \mathcal{D}_{\mathrm{KL}}(\mathbb{P}_{\mathrm{paths}} \| \mathbb{Q}_{\mathrm{paths}}^{T \to 0}) - \mathcal{D}_{\mathrm{KL}}(\mathbb{P}_{\mathrm{paths}} \| \mathbb{Q}_{\mathrm{paths}}^{T \to t \to 0}),$$

Thus, by maximizing the divergence on the left-hand side, we effectively minimize the discrepancy $\mathcal{D}_{\mathrm{KL}}(\mathbb{P}_{\mathrm{paths}} \| \mathbb{Q}_{\mathrm{paths}}^{T \to t \to 0})$ between the true distribution $\mathbb{P}_{\mathrm{paths}}$ and the optimized schedule $\mathbb{Q}_{\mathrm{paths}}^{T \to t \to 0}$.

Based on this, we can rewrite our optimization objective as follows (Proof in Appendix B.4):

$$t^* = \underset{t \in (T,0)}{\arg\max} \, \text{KLUB}_{\mathbb{Q}_T}(\mathbb{Q}_0^{T \to t \to 0} \| \mathbb{Q}_0^{T \to 0}) = \underset{t \in (T,0)}{\arg\max} \, \mathbb{E}_{\mathbb{Q}_{\text{paths}}^{T \to t \to 0}} \left[ \sum_{i \neq j} \log \frac{R_t(i,j)}{R_T(i,j)} \sum_{0 < u \leq t} H_u^{ij} \right] \quad (13)$$

Equation Eq. (13) offers a significant computational advantage over Eq. (8) by eliminating the need to compute the rate matrix at every transition time. Note that, calculating the reverse rate matrix involves neural network evaluations. Given a trajectory $(X_u)_{u \in [T,0]}$, Eq. (8) requires computing the rate matrix $R_u$ for all transitions where $H_u^{ij} = 1$. In contrast, Eq. (13) decouples the rate matrix from the transition times $u$, requiring computations only at times $T$ and $t$. Moreover, between times $T$ and $t$, the rate matrices for both $\mathbb{Q}_{\text{paths}}^{T \to t \to 0}$ and $\mathbb{Q}_{\text{paths}}^{T \to 0}$ are equal to $R_T$, eliminating the need for additional computations in this interval.

**Technique 2: Applying the Law of Total Expectation** Instead of directly compute the expectation over $\mathbb{Q}_{\text{paths}}^{T \to t \to 0}$, we compute expectation of conditional expectation, using the law of total expectation, i.e., $\mathbb{E}[X] = \mathbb{E}[\mathbb{E}[X|Y]]$:

$$\mathbb{E}_{\mathbb{Q}_{\text{paths}}^{T \to t \to 0}} \left[ \sum_{i \neq j} \log \frac{R_t(i,j)}{R_T(i,j)} \sum_{0 < u \leq t} H_u^{ij} \right] = \mathbb{E}_{\mathbb{Q}_{\text{paths}}^{T \to t}} \left[ \mathbb{E}_{\mathbb{Q}_{\text{paths}}^{t \to 0}} \left[ \sum_{i \neq j} \log \frac{R_t(i,j)}{R_T(i,j)} \sum_{0 < u \leq t} H_u^{ij} \middle| X_t = i \right] \right]$$

On the left side, to compute the expected value, we would need to sample the entire trajectory $(X_u)_{u \in [T,0]}$ and then calculate the inner sum. On the right side, this process is broken down into two steps: first, we sample a trajectory from $X_T$ to $X_t$, and then, given $X_t$, we sample from $X_t$ to $X_0$ to compute the inner expectation. Our main insight is that the expected value in the second step can be obtained in closed form.

The term $\sum_{0 < u \leq t} H_u^{ij}$ counts the number of transitions from state $i$ to state $j$ in the interval $[t,0)$. Since $\mathbb{Q}_{\text{paths}}^{t \to 0}$ uses a single rate matrix over this interval, for $i \neq j$, the expected number of transitions from $i$ to $j$ is approximately given by (Campbell et al., 2022, Sec. 4.3):

$$\mathbb{E}_{\mathbb{Q}_{\text{paths}}^{t \to 0}} \left[ \sum_{0 < u \leq t} H_u^{ij} \right] \approx R_t(i,j)\Delta t \quad (14)$$

where $\Delta t$ represents the size of the time interval.

Using this result, we can simplify Equation Eq. (13) as follows (Derivation in Appendix B.5):

$$\text{KLUB}_{\mathbb{Q}_T} \left( \mathbb{Q}_0^{T \to t \to 0} \, \| \, \mathbb{Q}_0^{T \to 0} \right) \approx \mathbb{E}_{\mathbb{Q}_{\text{paths}}^{T \to t}} \left[ \sum_{j \neq X_t} \log \frac{R_t(X_t, j)}{R_T(X_t, j)} \times R_t(X_t, j)\Delta t \right]. \quad (15)$$

In Eq. (13), obtaining a reliable KLUB estimation requires sampling trajectories to capture transitions between various states $i$ and $j$, which is both inaccurate and sample-inefficient—especially when dealing with large state spaces. For instance, in text generation tasks where the state space can be around 50,257, it's practically impossible to estimate the transition ratios between all pairs $(i, j)$ through sampling alone. In contrast, Eq. (15) allows us to compute this component in closed form, leading to more reliable and efficient calculations. The full algorithm for KLUB computation, combining Techniques 1 and 2, is provided in Appendix C.1.

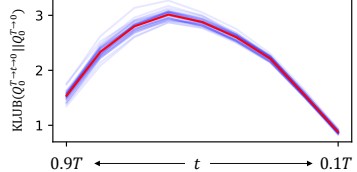

Figure 4: The values of KLUB with respect to $t$. Blue lines show estimated results from individual $(X_t)_{t \in [T,0]}$, while the red line is the average.

Now, we can maximize Eq. (15) using a standard optimization algorithm. Before selecting the algorithm, we conduct a preliminary check to observe how the KLUB value changes with respect to $t$ (Figure 4). Interestingly, it exhibits a unimodal shape. Since we are working with a

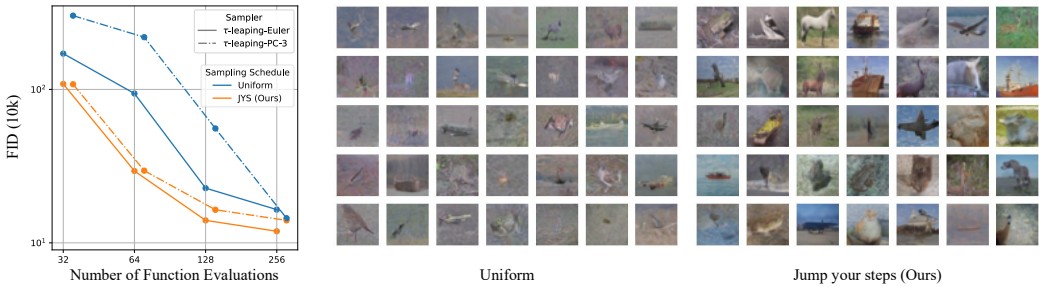

Figure 6: *(Left)* Performance comparisons on CIFAR-10. *(Right)* Samples generated using the uniform and JYS schedules, both with NFE 64.

single variable $t$ in the unimodal optimization landscape, we use the golden section search (Press, 2007), a well-known one-dimensional search algorithm, to find the value of $t$ that maximizes KLUB. This method has the advantage of not relying on hyperparameters like learning rate, which can significantly affect performance.

## 4 EXPERIMENTS

In this section, we evaluate the *Jump Your Steps* (JYS) sampling schedule across various datasets and models. We compare the JYS schedule with the uniform sampling schedule, which sets all intervals to the same size. Except for the Countdown dataset, we use open-sourced pretrained models for our experiments. It is important to note that the Gillespie algorithm is only applicable to an absorbing transition matrix, as uniform or Gaussian transition kernels do not have a fixed number of transitions. For further experimental details and additional qualitative results, please refer to the Appendix D and E.6.

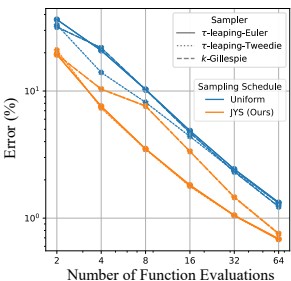

Figure 5: Performance comparisons on Countdown.

### 4.1 THE COUNTDOWN DATASET

Following Zhao et al. (2024), to evaluate our sampling schedule performance, we created a synthetic sequence dataset with a strong position-wise correlation structure. Each sample consists of 256 tokens, and each token has a value between 0 and 31. Each data sequence $X^{0:255}$ is generated according to the following rules:

$$X^0 \sim \text{Uniform}\{1, \ldots, S\},$$

$$X^{d+1} \mid X^d \sim \begin{cases} \delta_{X^d-1} & \text{if } X^d \neq 0 \\ \text{Uniform}\{1, \ldots, S\} & \text{if } X^d = 0 \end{cases}$$

We trained a SEDD (Lou et al., 2024) with an absorb transition matrix on this generated data. We measure the model performance by the proportion of generated samples that violated the rule, i.e., failed to count downwards from the previous token. The results are shown in Figure 5. We observe that the JYS schedule has fewer errors compared to the uniform schedule for the same NFE.

### 4.2 CIFAR-10

We demonstrate our sampling schedule in the image domain. For this experiment, we use a pretrained model from Campbell et al. (2022), which employs a gaussian transition matrix and denoising parameterization. Each data sample is a flattened image with a length of $3 \times 32 \times 32$, composed of tokens with values ranging from 0 to 255.

Figure 6 (left) shows the FID score using 50k samples with the number of function evaluations (NFE) from 32 to 256. We observe that, for all NFEs, the JYS schedule yields a better FID score

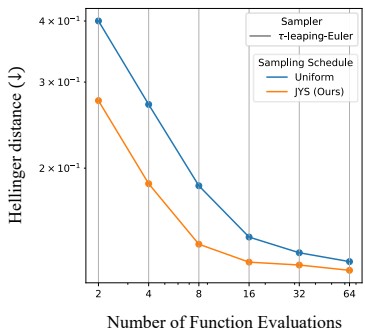

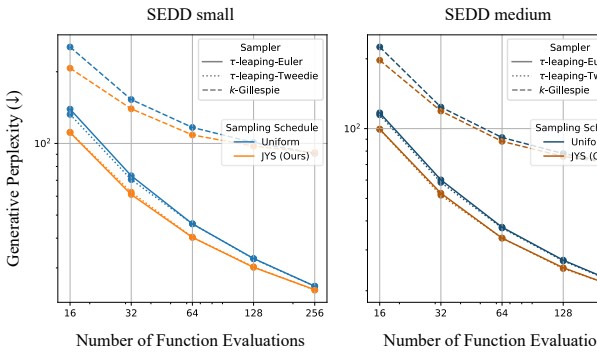

Figure 7: Performance comparisons on piano note generation.

Figure 8: Performance comparisons on text generation. Generative perplexity is measured by using GPT-2-large.

at the same NFE compared to the uniform schedule. Figure 6 (right) shows randomly generated unconditional CIFAR-10 samples with NFE = 64. The uniform schedule produces blurry images, whereas the images generated using the JYS schedule exhibit clearer colors and shapes of objects.

### 4.3 PIANO NOTE

We test our method on conditional piano note generation using the Lakh pianoroll dataset (Raffel, 2016; Dong et al., 2018). For this experiment, we employ a pretrained model from Campbell et al. (2022), which uses a uniform transition matrix and denoising parameterization. Each data sequence contains 256 timesteps (16 per bar), and we measure performance by conditioning on two bars to generate the remaining 14 bars, following the setup in Campbell et al. (2022).

We evaluate how different the generated results were when using a smaller NFE (from 2 to 64), compare to samples generated with an NFE of 512. Specifically, we calculate the Hellinger distance between the note distributions in the generated samples. The results are presented in Figure 7. Given the same NFE, we observe that samples generated using our method were more similar to those generated with a high NFE.

### 4.4 TEXT MODELING

Finally, we validate our method on text generation. For this experiment, we use a pretrained model from Lou et al. (2024), which employs an absorbing transition matrix and score-based parameterization. We use two model sizes, SEDD-small and SEDD-medium, in the experiments; both models use the GPT-2 tokenizer and were trained on OpenWebText. The JYS schedule, optimized on SEDD-small, is also used for the experiments with SEDD-medium.

Following Lou et al. (2024), we measure the generative perplexity of sampled sequences (using a GPT-2 large for evaluation). We generated 1,024 samples and each sample constructed with sequences of 1,024 tokens. We simulate 16 to 256 NFE for generation. Figure 8 shows the results, demonstrating better perplexity at the same NFE.

### 4.5 CHARACTERISTICS OF JUMP-YOUR-STEP SAMPLING SCHEDULE

In Section 3.1, we hypothesized that in regions where the conditional mutual information is low, the CDE would also be small, allowing steps to be skipped with minimal performance degradation. Here, we aim to verify if the JYS operates as expected according to this hypothesis.

Figure 9 shows the JYS sampling schedules optimized for various transition matrices. First, in the *absorb* case (Left), as discussed in Figure 1 (Bottom), we observe that large intervals are concentrated toward the latter part of the process. This occurs because the previously generated tokens help reduce the uncertainty of other tokens. In contrast, in the *uniform* case (Middle), large intervals appear at the beginning. This can be understood as a result of $X_t$ following a uniform distribution, making the tokens independent, leading to lower mutual information for larger $t$. Lastly, for

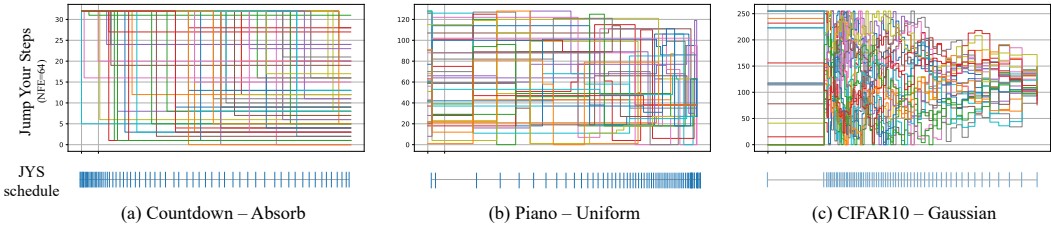

Figure 9: Sampling trajectories for different dataset-transition matrix combinations. *(Top)* Trajectories using the JYS schedule. *(Bottom)* Optimized sampling schedule with JYS.

the *Gaussian* transition (Right), large intervals appear initially and then increase again over time. This pattern suggests that, initially, tokens behave independently like in the uniform case, but after a certain timestep, the effect of resolving uncertainty, similar to the absorb case, becomes more significant as more tokens are generated.

## 5 RELATED WORK

**Efficient Sampling for Continuous Diffusion Models**  After the seminal work by Song et al. (2021b), which interpreted diffusion models as a Stochastic or Ordinary Differential Equations (SDE/ODE), various SDE (Jolicoeur-Martineau et al., 2021; Xu et al., 2023) and ODE solvers (Song et al., 2021a; Lu et al., 2022; Zhang & Chen, 2023; Dockhorn et al., 2022; Liu et al., 2022; Zheng et al., 2023) have been proposed to improve sampling speed.

The work most closely related to ours is "Align Your Step" (Sabour et al., 2024) and AdaptiveSchedules (Chen et al.), which focuses on sampling schedule optimization in continuous diffusion models. In contrast, our approach targets DDMs, where we derive the KLUB for CTMCs to optimize the sampling schedule. We also propose a computationally efficient algorithm for KLUB computation and optimization.

**Discrete Diffusion Models**  Several approaches have been developed for training DDMs, including denoising parameterization (Austin et al., 2021; Campbell et al., 2022; Gu et al., 2022; Gat et al., 2024; Campbell et al., 2024; Shi et al., 2024; Sahoo et al., 2024) and score parameterization (Sun et al., 2023; Meng et al., 2022; Lou et al., 2024). Types of DDMs that do not fall into these two categories are also being continuously proposed (Hoogeboom et al., 2021b;a; Santos et al., 2023). Recently, SEDD has outperformed GPT-2 in text modeling, gaining traction as an alternative to autoregressive models (Deschenaux & Gulcehre, 2024).

In terms of sampling, two main directions have emerged. The first focuses on efficient sampling, with $\tau$-leaping for CTMC and methods like analytic sampling (Sun et al., 2023), Tweedie sampling (Lou et al., 2024), and $k$-Gillespie (Zhao et al., 2024) improving accuracy. The second aims to reduce compounding error via corrector steps, such as random correctors (Campbell et al., 2022), separate corrector training (Lezama et al., 2022), or enabling the model to act as an informed corrector (Zhao et al., 2024). These methods complement the sampling schedule optimization explored in this paper and can be used together for further improvements. Recently, as a concurrent work, DNDM (Chen et al., 2024) proposed accelerated sampling through a non-Markovian framework that uses token transition times sampled from a beta distribution.

## 6 CONCLUSIONS

We present *Jump Your Steps*, a principled method designed to optimize the sampling schedule and minimize these numerical errors without incurring additional computational costs during inference. Unlike existing approaches that rely on extra computational efforts, such as predictor-corrector methods, our technique operates independently and efficiently. Through extensive evaluations on synthetic and real-world datasets—including monophonic piano, image, and text generation—we demonstrate that our method consistently enhances performance across different transition kernels in DDMs and effectively complements various samplers.

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

# Appendix

## A  DISCUSSIONS

### A.1  LIMITATIONS

Although the JYS sampling schedule is a method that can improve sampling quality in a plug-and-play manner for general DDMs, there are still problems that need to be solved. The first drawback is that, due to the nature of the hierarchical breakdown strategy, we can only optimize sampling schedules where NFE = $2^K$, so flexibility is reduced. While we propose a method to select an appropriate subset of the sampling schedule when the number of function evaluations (NFE) is not equal to $2^K$, optimizing the sampling schedule for arbitrary NFE using a principled approach is an interesting direction for future study.

Another theoretical limitation that needs to be addressed is that technique 2 uses the single-state transition assumption (Eq. 14)). This assumption is completely valid for the absorb transition kernel but does not apply to other transition kernels. This could potentially make the sampling schedule optimization suboptimal. Addressing this issue is interesting future work.

### A.2  DISCUSSIONS

**What does it mean to ignore the accumulated error from the previous steps?**   The "accumulated error from the previous steps that affects consecutive steps" occurs when a generated $x_t$ does not belong to the distribution $q(x_t)$, i.e., generated dataset mismatched with the training dataset. We refer to this error as *exposure bias* [1]. While exposure bias is indeed an interesting problem to address in generative model sampling, our work focuses specifically on resolving the CDE caused by

parallel decoding. Therefore, we chose to set aside exposure bias in this study. To improve clarity, we have added a discussion of this point to Section X of the revised paper.

**Optimization using gradient descent** As we mentioned in the main text, we perform optimization using the golden-section search algorithm. In preliminary experiments, we also tried the gradient descent algorithm but encountered two issues. The first was that gradient estimation was very noisy, and the second was that it was too sensitive to hyperparameters, i.e., the learning rate. Particularly, the second drawback made it impractical to use because we had to perform hyperparameter search optimization every time we optimized each step.

**Is there an alternative optimization algorithm to the hierarchical breakdown?** As part of our preliminary experiments, we explored an alternative strategy—hierarchical merging—where smaller, fragmented sampling schedules are combined into a coarse sampling schedule. However, our experimental results indicated that this approach often gets stuck in local optima and fails to deliver significant performance improvements. We will include this discussion in the revised version of the paper.

## B THEORETICAL DETAILS

### B.1 PROOF OF THEOREM 3.1

Although we treated the two-dimensional case where we have $X_t = (X_t^1, X_t^2)$ in the main body, we consider the general $d$-dimensional case with $X_t = (X_t^1, \ldots, X_t^d)$. In that case, the definition of $\mathcal{E}_{\text{CDE}}$ is given by

$$\mathcal{E}_{\text{CDE}}(s \to t | \mathbf{x}_s) \triangleq \mathcal{D}_{\text{KL}}(P_{\mathbf{x}_t | \mathbf{x}_s} \| P_{X_t^1 | \mathbf{x}_s} \otimes \cdots \otimes P_{X_t^d | \mathbf{x}_s}). \tag{16}$$

In general, for a discrete probability distribution $\mathbb{P}_{\text{prior}}(\cdot)$ over the space $\mathcal{S}$ and a conditional distribution (or denoiser) $\mathbb{P}_{\text{cond}}(\cdot|\cdot)$ over the same space, i,e, $\mathbb{P}_{\text{cond}} : \mathcal{S} \times \mathcal{S} \to \mathbb{R}$, let us just write the resulting distribution as follows:

$$\mathbb{P}_{\text{cond}}\mathbb{P}_{\text{prior}} \triangleq \mathbb{E}_{\mathbf{x} \sim \mathbb{P}_{\text{prior}}}[\mathbb{P}_{\text{cond}}(\cdot|\mathbf{x})] = \sum_{x \in \mathcal{S}} \mathbb{P}_{\text{cond}}(\cdot|x)\mathbb{P}_{\text{prior}}(x).$$

Then, if we define $\mathbb{P}_{t_{i+1}|t_i}(\cdot|\mathbf{x}_{t_i}) \triangleq P_{X_{t_{i+1}}|\mathbf{x}_{t_i}}$ and $\mathbb{Q}_{t_{i+1}|t_i}(\cdot|\mathbf{x}_{t_i}) \triangleq P_{X_t^1|\mathbf{x}_s} \otimes \cdots \otimes P_{X_t^d|\mathbf{x}_s}$ following equation 16, we can denote the target distributions in the theorem as follows:

$$\mathbb{P}_0 = \mathbb{P}_{t_N|t_{N-1}} \cdots \mathbb{P}_{t_1|t_0}\mathbb{P}_{t_0}, \qquad \mathbb{Q}_0 \triangleq \mathbb{Q}_0^{T \to t_1 \to \cdots \to 0} = \mathbb{Q}_{t_N|t_{N-1}} \cdots \mathbb{Q}_{t_1|t_0}\mathbb{P}_{t_0}.$$

For simplicity, let us also define the mid-time distributions as

$$\mathbb{P}_{t_i} \triangleq \mathbb{P}_{t_i|t_{i-1}} \cdots \mathbb{P}_{t_1|t_0}\mathbb{P}_{t_0}, \qquad \mathbb{Q}_{t_i} \triangleq \mathbb{Q}_{t_i|t_{i-1}} \cdots \mathbb{Q}_{t_1|t_0}\mathbb{P}_{t_0}, \qquad \mathbb{Q}_{t_0} \triangleq \mathbb{P}_{t_0}.$$

In the proof of the theorem, we use the following well-known lemma:

**Lemma B.1.** *For discrete finite sets $\mathcal{X}, \mathcal{Y}$ and the joint probability distributions $\mathbb{P}_{X,Y}$ and $\mathbb{Q}_{X,Y}$ on $\mathcal{X} \times \mathcal{Y}$, we have*

$$\mathcal{D}_{\text{KL}}(\mathbb{P}_{X,Y}\|\mathbb{Q}_{X,Y}) = \mathcal{D}_{\text{KL}}(\mathbb{P}_X\|\mathbb{Q}_X) + \mathbb{E}_{x \sim \mathbb{P}_X}\left[\mathcal{D}_{\text{KL}}(\mathbb{P}_{Y|X}(\cdot|x)\|\mathbb{Q}_{Y|X}(\cdot|x))\right],$$

*where $\mathbb{P}_X, \mathbb{Q}_X$ and $\mathbb{P}_{Y|X}, \mathbb{Q}_{Y|X}$ respectively denote the marginal and conditional distributions.*

*Proof.* We can just compute as

$$\mathcal{D}_{\text{KL}}(\mathbb{P}_{X,Y}\|\mathbb{Q}_{X,Y}) = \sum_{y,x} \mathbb{P}_{Y|X}(y|x)\mathbb{P}_X(x) \log \frac{\mathbb{P}_{Y|X}(y|x)\mathbb{P}_X(x)}{\mathbb{Q}_{Y|X}(y|x)\mathbb{Q}_X(x)}$$

$$= \sum_{y,x} \mathbb{P}_{Y|X}(y|x)\mathbb{P}_X(x) \log \frac{\mathbb{P}_X(x)}{\mathbb{Q}_X(x)} + \sum_{y,x} \mathbb{P}_{Y|X}(y|x)\mathbb{P}_X(x) \log \frac{\mathbb{P}_{Y|X}(y|x)}{\mathbb{Q}_{Y|X}(y|x)}$$

$$= \mathcal{D}_{\text{KL}}(\mathbb{P}_X\|\mathbb{Q}_X) + \sum_x \mathbb{P}_X(x)\mathcal{D}_{\text{KL}}(\mathbb{P}_{Y|X}(\cdot|x)\|\mathbb{Q}_{Y|X}(\cdot|x)),$$

which leads to the stated equation. □

Now we can proceed to the proof of the theorem.

*Proof of Theorem 3.1.* Let us now consider the case where $s = t_{i-1}$ and $t = t_i$ for some $i > 0$ specifically. Let

$$\mathbb{P}_{t,s}(y,x) \triangleq \mathbb{P}_{t|s}(y|x)\mathbb{P}_s(x), \qquad \mathbb{Q}_{t,s}(y,x) \triangleq \mathbb{Q}_{t|s}(y|x)\mathbb{Q}_s(x)$$

to denote joint distributions on $\mathcal{S} \times \mathcal{S}$. Then, since $\mathbb{P}_t = \mathbb{P}_{t|s}\mathbb{P}_s$ and $\mathbb{Q}_t = \mathbb{Q}_{t|s}\mathbb{Q}_s$ are marginal distributions of $\mathbb{P}_{t,s}$ and $\mathbb{Q}_{t,s}$, we have (from Lemma B.1) the following:

$$\begin{aligned}
\mathcal{D}_{\mathrm{KL}}(\mathbb{P}_t\|\mathbb{Q}_t) &\leq \mathcal{D}_{\mathrm{KL}}(\mathbb{P}_{t,s}\|\mathbb{Q}_{t,s}) \\
&= \mathcal{D}_{\mathrm{KL}}(\mathbb{P}_s\|\mathbb{Q}_s) + \mathbb{E}_{x\sim\mathbb{P}_s}[\mathcal{D}_{\mathrm{KL}}(\mathbb{P}_{t|s}(\cdot|x)\|\mathbb{Q}_{t|s}(\cdot|x))] \\
&= \mathcal{D}_{\mathrm{KL}}(\mathbb{P}_s\|\mathbb{Q}_s) + \mathbb{E}_{\mathbf{x}_s\sim\mathbb{P}_s}[\mathcal{E}_{\mathrm{CDE}}(s\to t|\mathbf{x}_s)] \\
&= \mathcal{D}_{\mathrm{KL}}(\mathbb{P}_s\|\mathbb{Q}_s) + \mathcal{E}_{\mathrm{CDE}}(s\to t).
\end{aligned} \tag{17}$$

From Lemma B.1, equality holds if and only if $\mathcal{D}_{\mathrm{KL}}(\mathbb{P}_{t|s}\|\mathbb{Q}_{t|s}) = 0$.

By iteratively using equation 17, we obtain

$$\begin{aligned}
\mathcal{D}_{\mathrm{KL}}(\mathbb{P}_0\|\mathbb{Q}_0) &= \mathcal{D}_{\mathrm{KL}}(\mathbb{P}_{t_N}\|\mathbb{Q}_{t_N}) \\
&\leq \mathcal{D}_{\mathrm{KL}}(\mathbb{P}_{t_{N-1}}\|\mathbb{Q}_{t_{N-1}}) + \mathcal{E}_{\mathrm{CDE}}(t_{N-1}\to t_N) \\
&\leq \mathcal{D}_{\mathrm{KL}}(\mathbb{P}_{t_{N-2}}\|\mathbb{Q}_{t_{N-2}}) + \mathcal{E}_{\mathrm{CDE}}(t_{N-2}\to t_{N-1}) + \mathcal{E}_{\mathrm{CDE}}(t_{N-1}\to t_N) \\
&\quad\vdots \\
&\leq \mathcal{D}_{\mathrm{KL}}(\mathbb{P}_{t_0}\|\mathbb{Q}_{t_0}) + \sum_{i=0}^{N-1}\mathcal{E}_{\mathrm{CDE}}(t_i\to t_{i+1}).
\end{aligned}$$

Since we set $\mathbb{P}_{t_0} = \mathbb{Q}_{t_0} = \pi$, where $\pi$ is terminal distribution, we have completed the proof. Furthermore, equality holds if and only if $\mathcal{D}_{\mathrm{KL}}(\mathbb{P}_{t_i|t_{i+1}}\|\mathbb{Q}_{t_i|t_{i+1}}) = 0$ for all $t \in \{0,\cdots,N-1\}$. □

### B.1.1 COMPARISON: NEGATIVE ELBO VS SUM OF CDEs

Let us just denote by $\mathbb{P}_{t_0,\ldots,t_i}$ the joint distribution of the variables $(\mathbf{x}_{t_0},\ldots,\mathbf{x}_{t_i})$ generated based on $\mathbb{P}_{t_0}$ and $\mathbb{P}_{t_{j+1}|t_j}$'s. Similarly define $\mathbb{Q}_{t_0,\ldots,t_i}$. We can also see the sum of CDEs as the KL divergence between discrete path measures. Indeed, by using Lemma B.1 with decomposing $(\mathbf{x}_{t_0},\ldots,\mathbf{x}_{t_{i+1}})$ into $(\mathbf{x}_{t_0},\ldots,\mathbf{x}_i)$ and $\mathbf{x}_{t_{i+1}}$, we have

$$\begin{aligned}
&\mathcal{D}_{\mathrm{KL}}(\mathbb{P}_{t_0,\ldots,t_{i+1}}\|\mathbb{Q}_{t_0,\ldots,t_{i+1}}) \\
&= \mathcal{D}_{\mathrm{KL}}(\mathbb{P}_{t_0,\ldots,t_i}\|\mathbb{Q}_{t_0,\ldots,t_i}) \\
&\quad + \mathbb{E}_{\mathbf{x}_{t_0},\ldots,\mathbf{x}_{t_i}\sim\mathbb{P}_{t_0,\ldots,t_i}}\left[\mathcal{D}_{\mathrm{KL}}(\mathbb{P}_{t_{i+1}|t_0,\ldots,t_i}(\cdot|\mathbf{x}_{t_0},\ldots,\mathbf{x}_{t_i})\|\mathbb{Q}_{t_{i+1}|t_0,\ldots,t_i}(\cdot|\mathbf{x}_{t_0},\ldots,\mathbf{x}_{t_i}))\right] \\
&= \mathcal{D}_{\mathrm{KL}}(\mathbb{P}_{t_0,\ldots,t_i}\|\mathbb{Q}_{t_0,\ldots,t_i}) + \mathbb{E}_{\mathbf{x}_{t_i}\sim\mathbb{P}_{t_i}}\left[\mathcal{D}_{\mathrm{KL}}(\mathbb{P}_{t_{i+1}|t_i}(\cdot|\mathbf{x}_{t_i})\|\mathbb{Q}_{t_{i+1}|t_i}(\cdot|\mathbf{x}_{t_i}))\right]
\end{aligned}$$

where the second equality comes from the Markov property of the defined processes. By iteratively using this, we have

$$\begin{aligned}
&\mathcal{D}_{\mathrm{KL}}(\mathbb{P}_{t_0,\ldots,t_N}\|\mathbb{Q}_{t_0,\ldots,t_N}) \\
&= \mathcal{D}_{\mathrm{KL}}(\mathbb{P}_{t_0}\|\mathbb{Q}_{t_0}) + \sum_{i=0}^{N-1}\mathbb{E}_{\mathbf{x}_{t_i}\sim\mathbb{P}_{t_i}}\left[\mathcal{D}_{\mathrm{KL}}(\mathbb{P}_{t_{i+1}|t_i}(\cdot|\mathbf{x}_{t_i})\|\mathbb{Q}_{t_{i+1}|t_i}(\cdot|\mathbf{x}_{t_i}))\right] \tag{18} \\
&= \mathcal{D}_{\mathrm{KL}}(\mathbb{P}_{t_0}\|\mathbb{Q}_{t_0}) + \sum_{i=0}^{N-1}\mathcal{E}_{\mathrm{CDE}}(t_i\to t_{i+1}).
\end{aligned}$$

Now, notice that it is quite similar to the usual negative ELBO loss (e.g., $\mathcal{L}_{\mathrm{DT}}$ in Campbell et al. (2022, page 3)) to compare the $\mathbb{P}$ (target) and $\mathbb{Q}$ (approximation) in training diffusion models. By replacing $\mathbb{P}_{t_{i+1}|t_i}(\cdot|\mathbf{x}_{t_i})$ with $\mathbb{P}_{t_{i+1}|t_i,0}(\cdot|\mathbf{x}_{t_i},\mathbf{x}_0)$ by conditioning on $\mathbf{x}_0 \sim \mathbb{P}_0 = \mathbb{P}_{t_N}$, we mostly recover the negative ELBO.

Let us proceed more formally. The negative ELBO we consider (Campbell et al., 2022; Sohl-Dickstein et al., 2015) can be written as:

$$\mathcal{L}_{\text{NELBO}}(\mathbb{P}, \mathbb{Q}) = \mathbb{E}_{\mathbf{x}_0 \sim \mathbb{P}_0}[\mathcal{L}_{\text{NELBO}}^{\mathbf{x}_0}(\mathbb{P}, \mathbb{Q})],$$

where

$$\mathcal{L}_{\text{NELBO}}^{x}(\mathbb{P}, \mathbb{Q}) = \mathcal{D}_{\text{KL}}(\mathbb{P}_{t_0|t_N}(\cdot|x)\|\mathbb{Q}_{t_0}) + \mathbb{E}_{\mathbf{x}_{t_{N-1}} \sim \mathbb{P}_{t_{N-1}|t_N}(\cdot|x)}[-\log \mathbb{Q}_{t_N|t_{N-1}}(x|\mathbf{x}_{t_{N-1}})]$$

$$+ \sum_{i=0}^{N-2} \mathbb{E}_{\mathbf{x}_{t_i} \sim \mathbb{P}_{t_i|t_N}(\cdot|x)}[\mathcal{D}_{\text{KL}}(\mathbb{P}_{t_{i+1}|t_i,t_N}(\cdot|\mathbf{x}_{t_i},x)\|\mathbb{Q}_{t_{i+1}|t_i}(\cdot|\mathbf{x}_{t_i}))].$$

Note that the negative ELBO and our summation of CDEs (equation 18) are both trying to upper-bound the target KL divergence $\mathcal{D}_{\text{KL}}(\mathbb{P}_0\|\mathbb{Q}_0)$. In the following, we shall prove $\mathcal{D}_{\text{KL}}(\mathbb{P}_{t_0,\ldots,t_N}\|\mathbb{Q}_{t_0,\ldots,t_N}) \leq \mathcal{L}_{\text{NELBO}}(\mathbb{P}, \mathbb{Q})$ to see our bound actually gives a tighter approximation of $\mathcal{D}_{\text{KL}}(\mathbb{P}_0\|\mathbb{Q}_0)$ than the negative ELBO computed at timesteps $\{T \to t_1 \to t_2 \to \ldots \to t_{N-1} \to 0\}$.

Since $\mathbb{P}_{t_N|t_{N-1},t_N}(\cdot|\mathbf{x}_{t_{N-1}}, x)$ is the delta distribution at $x$, we can formally rewrite a term in $\mathcal{L}_{\text{NELBO}}^{x}$ as $-\log \mathbb{Q}_{t_N|t_{N-1}}(x|\mathbf{x}_{t_{N-1}}) = \mathcal{D}_{\text{KL}}(\mathbb{P}_{t_N|t_{N-1},t_N}(\cdot|\mathbf{x}_{t_{N-1}}, x)\|\mathbb{Q}_{t_N|t_{N-1}}(\cdot|\mathbf{x}_{t_{N-1}}))$ and so obtain

$$\mathcal{L}_{\text{NELBO}}^{x}(\mathbb{P}, \mathbb{Q}) = \mathcal{D}_{\text{KL}}(\mathbb{P}_{t_0|t_N}(\cdot|x)\|\mathbb{Q}_{t_0})$$

$$+ \sum_{i=0}^{N-1} \mathbb{E}_{\mathbf{x}_{t_i} \sim \mathbb{P}_{t_i|t_N}(\cdot|x)}[\mathcal{D}_{\text{KL}}(\mathbb{P}_{t_{i+1}|t_i,t_N}(\cdot|\mathbf{x}_{t_i},x)\|\mathbb{Q}_{t_{i+1}|t_i}(\cdot|\mathbf{x}_{t_i}))]. \quad (19)$$

By using equation 18 and equation 19, we prove $\mathcal{D}_{\text{KL}}(\mathbb{P}_{t_0,\ldots,t_N}\|\mathbb{Q}_{t_0,\ldots,t_N}) \leq \mathcal{L}_{\text{NELBO}}(\mathbb{P}, \mathbb{Q}) = \mathbb{E}_{\mathbf{x}_0 \sim \mathbb{P}_0}[\mathcal{L}_{\text{NELBO}}^{\mathbf{x}_0}(\mathbb{P}, \mathbb{Q})]$ in a term-by-term manner. We shall heavily use the convexity of KL divergence (Cover & Thomas, 2006, Theorem 2.7.2) in the following form:

$$\mathcal{D}_{\text{KL}}(\mathbb{E}_{x \sim \pi}[\mathbb{P}_{Y|X}(\cdot|x)]\|\mathbb{Q}_Y) \leq \mathbb{E}_{x \sim \pi}[\mathcal{D}_{\text{KL}}(\mathbb{P}_{Y|X}(\cdot|x)\|\mathbb{Q}_Y)], \quad (20)$$

where $\mathbb{P}_{Y|X}$ is a conditional distribution and $\pi$ is any reference probability distribution.

**First term.** We can prove

$$\mathcal{D}_{\text{KL}}(\mathbb{P}_{t_0}\|\mathbb{Q}_{t_0}) \leq \mathbb{E}_{x \sim \mathbb{P}_{t_N}}[\mathcal{D}_{\text{KL}}(\mathbb{P}_{t_0|t_N}(\cdot|x)\|\mathbb{Q}_{t_0})] \quad (21)$$

first. Since we have $\mathbb{E}_{x \sim \mathbb{P}_{t_N}}[\mathbb{P}_{t_0|t_N}(\cdot|x)] = \mathbb{P}_{t_0}$, it directly follows from the convexity (equation 20).

**Summation term.** For each term in the summation, we can also prove the following:

$$\mathbb{E}_{\mathbf{x}_{t_i} \sim \mathbb{P}_{t_i}}[\mathcal{D}_{\text{KL}}(\mathbb{P}_{t_{i+1}|t_i}(\cdot|\mathbf{x}_{t_i})\|\mathbb{Q}_{t_{i+1}|t_i}(\cdot|\mathbf{x}_{t_i}))]$$

$$\leq \mathbb{E}_{x \sim \mathbb{P}_{t_N}}\mathbb{E}_{\mathbf{x}_{t_i} \sim \mathbb{P}_{t_i|t_N}(\cdot|x)}[\mathcal{D}_{\text{KL}}(\mathbb{P}_{t_{i+1}|t_i,t_N}(\cdot|\mathbf{x}_{t_i},x)\|\mathbb{Q}_{t_{i+1}|t_i}(\cdot|\mathbf{x}_{t_i}))]. \quad (22)$$

Indeed, we can replace $\mathbb{E}_{x \sim \mathbb{P}_{t_N}}\mathbb{E}_{\mathbf{x}_{t_i} \sim \mathbb{P}_{t_i|t_N}(\cdot|x)}$ with $\mathbb{E}_{\mathbf{x}_{t_i} \sim \mathbb{P}_{t_i}}\mathbb{E}_{x \sim \mathbb{P}_{t_N|t_i}(\cdot|\mathbf{x}_{t_i})}$ and it thus suffices to prove

$$\mathcal{D}_{\text{KL}}(\mathbb{P}_{t_{i+1}|t_i}(\cdot|\mathbf{x}_{t_i})\|\mathbb{Q}_{t_{i+1}|t_i}(\cdot|\mathbf{x}_{t_i})) \leq \mathbb{E}_{x \sim \mathbb{P}_{t_N|t_i}(\cdot|\mathbf{x}_{t_i})}[\mathcal{D}_{\text{KL}}(\mathbb{P}_{t_{i+1}|t_i,t_N}(\cdot|\mathbf{x}_{t_i},x)\|\mathbb{Q}_{t_{i+1}|t_i}(\cdot|\mathbf{x}_{t_i}))]$$

for each $\mathbf{x}_{t_i}$. Since $\mathbb{E}_{x \sim \mathbb{P}_{t_N|t_i}(\cdot|\mathbf{x}_{t_i})}[\mathbb{P}_{t_{i+1}|t_i,t_N}(\cdot|\mathbf{x}_{t_i},x)] = \mathbb{P}_{t_{i+1}|t_i}(\cdot|\mathbf{x}_{t_i})$, this again follows from the convexity.

By combining equation 21 and equation 22, we obtain $\mathcal{D}_{\text{KL}}(\mathbb{P}_{t_0,\ldots,t_N}\|\mathbb{Q}_{t_0,\ldots,t_N}) \leq \mathcal{L}_{\text{NELBO}}(\mathbb{P}, \mathbb{Q})$, which means the CDE-based upper bound is at most as loose as the commonly used negative ELBO bound. We can also come to this conclusion directly applying the convexity after showing $\mathcal{L}_{\text{NELBO}}(\mathbb{P}, \mathbb{Q}) = \mathbb{E}_{x \sim \mathbb{P}_0}[\mathcal{D}_{\text{KL}}(\mathbb{P}_{t_0,\ldots,t_{N-1},t_N|t_N}\|\mathbb{Q}_{t_0,\ldots,t_N})]$, which is also written in Campbell et al. (2022, page 3), but we chose the above term-by-term derivation since it would give additional intuitions.

### B.2 PROOF OF THEOREM 3.2

Although we state the theorem for a backward CTMC from time $s$ to time $t$ $(t < s)$ and bound the KL-divergence at time $t$, here we will just consider the forward CTMC from time $0$ to time $T$ and bound the KL-divergence at time $T$ for simplicity. After this change of the time direction and interval, the formal statement and proof of Theorem 3.2 is given in Theroem B.4.

To derive the KL-divergence upper bound (KLUB) for continuous-time Markov chains (CTMCs), we first adopt the change of measure $\frac{d\mathbb{P}_{\text{paths}}}{d\mathbb{Q}_{\text{paths}}}$ for CTMCs from Section 3 of Ding & Ning (2021). Next, we compute and organize the equation for the $D_{\text{KL}}(\mathbb{P}_{\text{paths}}|\mathbb{Q}_{\text{paths}}) = \mathbb{E}_{\mathbb{P}_{\text{paths}}}\left[\log \frac{d\mathbb{P}_{\text{paths}}}{d\mathbb{Q}_{\text{paths}}}\right]$.

We consider the following two forward CTMCs over $[0, T]$:

$$\begin{cases} \text{CTMC 1}: & q^1_{u+du|u}(y \mid x) = \delta_{xy} + R^1_t(x, y)du + o(du), \\ \text{CTMC 2}: & q^2_{u+du|u}(y \mid x) = \delta_{xy} + R^2_t(x, y)du + o(du). \end{cases}$$

Here, $R^1_t$ and $R^2_t$ represent the rate matrices of each CTMC, with a finite state space $S = \{x_1, \cdots, x_N\}$, and $du > 0$.

We introduce some notations. Define the functions $H^i_t, H^{ij}_t$ as follows:

$$H^i_t := \delta(X_t - x_i), \quad H^{ij}_t := H^j_t H^i_{t-},$$

where $\delta(\cdot)$ denotes the Dirac delta function, and $t-$ is the left limit of $t$. By definition, $H^{ij}_t = 1$ indicates a transition from $x_i$ to $x_j$ at time $t$. The CTMC $(X_t)_{t\in[0,T]}$ is defined as a function from the sample space $\Omega$ to the path space $\mathcal{C} \triangleq [0, T] \times S$, i.e., $X : \Omega \to \mathcal{C}$.

We define two probability measures over the path space:

- $\mathbb{P}_{\text{paths}}$, under which $(X^1_t)_{t\in[0,T]}$ has the law of CTMC 1.
- $\mathbb{Q}_{\text{paths}}$, under which $(X^2_t)_{t\in[0,T]}$ has the law of CTMC 2.

First, we adopt the change of measure for CTMCs from Ding & Ning (2021).

**Proposition B.2.** *(Ding & Ning (2021), Eq. 3.2) Consider a family of bounded real-valued processes $\{\kappa_t(i, j)\}_{i,j\in\{1,\cdots,N\}}$, such that $\kappa_t(i, j) > -1$ and $\kappa_{ii}(t) = 0$. Define $(\eta_t)_{0\le t\le T}$ as*

$$\eta_t = e^{-L_t} \prod_{0 < u \le T} \left(1 + \sum_{i,j=1}^N \kappa_u(i, j) H^{ij}_u\right),$$

*where $L_t = \int_0^t \sum_{i,j=1}^N \kappa_u(i, j) R^1_u(i, j) H^i_u du$. This result implies the existence of a probability measure $\mathbb{Q}$ defined by*

$$\frac{d\mathbb{Q}}{d\mathbb{P}} = \eta_t. \tag{23}$$

The following result allows us to define $\kappa_t(i, j)$ for two CTMCs.

**Proposition B.3.** *(Ding & Ning (2021), Eq. 3.4) For the probability measure $\mathbb{Q}$ defined in Eq.(23), if $(X^1_t)_{t\in[0,T]}$ is a CTMC under $\mathbb{P}$ with rate matrix $R^1$ and $(X^2_t)_{t\in[0,T]}$ is a CTMC under $\mathbb{Q}$ with rate matrix $R^2$, then $R^2$ satisfies*

$$R^2_{ii}(t) = -\sum_{j\neq i} R^2_t(i, j),$$
$$R^2_t(i, j) = (1 + \kappa_t(i, j))R^1_t(i, j).$$

From Proposition A.2, defining $\kappa_t(i, j) = R^2_t(i, j)/R^1_t(i, j) - 1$ (as in Proposition A.1), we can compute the change of measure between the two CTMC-defined measures $\mathbb{P}_{\text{paths}}$ and $\mathbb{Q}_{\text{paths}}$ as $\frac{d\mathbb{P}_{\text{paths}}}{d\mathbb{Q}_{\text{paths}}}$.

Now, we are ready to derive the KLUB for CTMCs:

**Theorem B.4.** (KL-divergence Upper bound, KLUB) *Consider the following two forward CTMCs:*

$$
\begin{cases}
CTMC\ 1: & q^1_{u+du|u}(y \mid x) = \delta_{xy} + R^1_t(x,y)du + o(du), \\
CTMC\ 2: & q^2_{u+du|u}(y \mid x) = \delta_{xy} + R^2_t(x,y)du + o(du).
\end{cases}
$$

*Here, $R^1_t$ and $R^2_t$ represent the rate matrices of each CTMC over $[0,T]$, with a finite state space $S = \{x_1, \cdots, x_N\}$. Let $\mathbb{P}_T$ and $\mathbb{Q}_T$ be the resulting probability distributions at the time $T$ of the outputs of CTMC 1 and 2, respectively. Then we have:*

$$
D_{\mathrm{KL}}(\mathbb{P}_T \| \mathbb{Q}_T) \leq \mathbb{E}_{\mathbb{P}_{\mathrm{paths}}} \left[ \sum_{\substack{H^{ij}_u = 1 \\ 0 < u \leq T}} \log \frac{R^1_u(i,j)}{R^2_u(i,j)} \cdot \right]
$$

*where, $\mathbb{P}_{\mathrm{paths}}$ refers to the distribution over path space $(X_t)_{t \in [0,T]} \in [0,T] \times S$ generated by running CTMC 1, and $H^{ij}_u = 1$ represent the transition of state from $x_i$ to $x_j$ at time $u$.*

*Proof.* This result is derived by combining Proposition A.1 and A.2 and rearranging the resulting expression.

$$
D_{\mathrm{KL}}\left(\mathbb{P}_{\mathrm{paths}} \| \mathbb{Q}_{\mathrm{paths}}\right) = \mathbb{E}_{\mathbb{P}_{\mathrm{paths}}} \left[ \log \frac{d\mathbb{P}_{\mathrm{paths}}}{d\mathbb{Q}_{\mathrm{paths}}} \right] \tag{24a}
$$

$$
= \mathbb{E}_{\mathbb{P}_{\mathrm{paths}}} \left[ \log \eta_t^{-1} \right] \tag{24b}
$$

$$
= \mathbb{E}_{\mathbb{P}_{\mathrm{paths}}} \left[ \int_0^t \sum_{i,j=1}^N \kappa_u(i,j) R^1_u(i,j) H^i_u du - \log \prod_{0 < u \leq T} \left( 1 + \sum_{i,j=1}^N \kappa_u(i,j) H^{ij}_u \right) \right] \tag{24c}
$$

$$
= \mathbb{E}_{\mathbb{P}_{\mathrm{paths}}} \left[ \int_0^t \sum_{i,j=1}^N (R^2_u(i,j) - R^1_u(i,j)) H^i_u du + \sum_{\substack{H^{ij}_u = 1 \\ 0 < u \leq T}} (\log R^1_u(i,j) - \log R^2_u(i,j)) \right] \tag{24d}
$$

$$
= \mathbb{E}_{\mathbb{P}_{\mathrm{paths}}} \left[ \sum_{\substack{H^{ij}_u = 1 \\ 0 < u \leq T}} (\log R^1_u(i,j) - \log R^2_u(i,j)) \right]. \tag{24e}
$$

Step (18c) follows from Proposition A.1, while step (18d) is derived by substituting $\kappa_t(i,j) = \frac{R^2_t(i,j)}{R^1_t(i,j)} - 1$ as defined in Proposition A.2. Additionally, step (18e) utilizes the property of the rate matrix, where $\sum_j R^1_t(i,j) = 0$ for any $t$ and $i$.

Finally, we denote the distributions generated by CTMC 1 and CTMC 2 at time $T$ as $\mathbb{P}_T$ and $\mathbb{Q}_T$, respectively. Since $\mathbb{P}_T$ and $\mathbb{Q}_T$ are the marginal distributions of $\mathbb{P}_{\mathrm{paths}}$ and $\mathbb{Q}_{\mathrm{paths}}$ at time $T$, by the data processing inequality, the KL-divergence $D_{\mathrm{KL}}(\mathbb{P}_T \| \mathbb{Q}_T)$ is upper-bounded by $D_{\mathrm{KL}}(\mathbb{P}_{\mathrm{paths}} \| \mathbb{Q}_{\mathrm{paths}})$, concluding the proof.

The equality holds if and only if $\lim_{dt \to 0} D_{KL}(\mathbb{P}_{t-dt|t} \| \mathbb{Q}_{t-dt|t}) = 0$, for all $t \in (0,T]$. To derive equality condition, we decompose $\mathbb{P}_{\mathrm{paths}}$ and use Lemma B.1 to get a bound.

$$
D_{KL}(\mathbb{P}_{\mathrm{paths}} \| \mathbb{Q}_{\mathrm{paths}}) = D_{KL}(\mathbb{P}_T \| \mathbb{Q}_T) + \mathbb{E}_{x \sim \mathbb{P}_T(x)} \left[ D_{KL}(\mathbb{P}_{0:T-dt|T} \| \mathbb{Q}_{0:T-dt|T}) \right] \tag{25}
$$

$$
= D_{KL}(\mathbb{P}_T \| \mathbb{Q}_T) + \mathbb{E}_{x \sim \mathbb{P}_T(x)} \left[ D_{KL}(\mathbb{P}_{T-dt|T} \| \mathbb{Q}_{T-dt|T}) \right] \tag{26}
$$

$$
+ \mathbb{E}_{x \sim \mathbb{P}_{T-dt}(x)} \left[ D_{KL}(\mathbb{P}_{0:T-2dt|T-dt} \| \mathbb{Q}_{0:T-2dt|T-dt}) \right] \tag{27}
$$

$$
= D_{KL}(\mathbb{P}_T \| \mathbb{Q}_T) + \sum_{t=dt}^T \mathbb{E}_{x \sim \mathbb{P}_t(x)} \left[ D_{KL}(\mathbb{P}_{t-dt|t} \| \mathbb{Q}_{t-dt|t}) \right] \tag{28}
$$

$\square$

It is important to note that the KLUB derived here for CTMCs does not fully capture the case of discrete diffusion models, where the reverse rate matrix depends on the state. Nonetheless, we believe the metric derived here provides a useful proxy for estimating the error introduced when using $\tau$-leaping to sample in discrete diffusion models. Finding KLUB for state-dependent CTMCs would be an interesting direction for future work.

### B.3 PROOF OF EQ. (9)

Let $X^i$ refers to the $i$-th dimension of $X$. Define $\mathbb{P}_{t_{i+1}|t_i}(\cdot|\mathbf{x}_{t_i}) \triangleq P_{X_{t_{i+1}}|\mathbf{x}_{t_i}}$ and $\mathbb{Q}_{t_{i+1}|t_i}(\cdot|\mathbf{x}_{t_i}) \triangleq P_{X_t^1|\mathbf{x}_s} \otimes \cdots \otimes P_{X_t^d|\mathbf{x}_s}$ We can now derive the following relationship:

$$\mathcal{E}_{\mathrm{CDE}}(s \to t \mid \mathbf{x}_s) \triangleq D_{KL}\left(\mathbb{P}_{t|s}\big\|\mathbb{Q}_{t|s}\right) \leq D_{KL}\left(\mathbb{P}_{[s,t]|s}\big\|\mathbb{Q}_{[s,t]|s}\right) = \mathrm{KLUB}_{\mathbf{x}_s}(\mathbb{P}_t\|\mathbb{Q}_t).$$

where $\mathrm{KLUB}_{\mathbf{x}_s}$ represents comparing two continuous-time Markov chains (CTMCs), both starting at the initial point $\mathbf{x}_s$.

Now we are ready to prove the result:

$$\begin{aligned}
\mathcal{E}_{\mathrm{CDE}}(s \to t) &\triangleq \mathbb{E}_{\mathbf{X}_s \sim \mathbb{P}_s}\left[\mathcal{E}_{\mathrm{CDE}}(s \to t \mid \mathbf{X}_s)\right] \\
&\leq \mathbb{E}_{\mathbf{X}_s \sim \mathbb{P}_s}\left[\mathrm{KLUB}_{\mathbf{X}_s}(\mathbb{P}_t\|\mathbb{Q}_t^{s \to t})\right] \\
&= \mathbb{E}_{\mathbf{X}_s \sim \mathbb{P}_s}\left[\mathbb{E}_{\mathbb{P}_{t|s}}\left[\sum_{\substack{H_u^{ij}=1 \\ 0 < u \leq T}} \log \frac{R_u^1(i,j)}{R_u^2(i,j)}\right]\right] \\
&= \mathbb{E}_{\mathbb{P}_{\mathrm{paths}}}\left[\sum_{\substack{H_u^{ij}=1 \\ 0 < u \leq T}} \log \frac{R_u^1(i,j)}{R_u^2(i,j)}\right] \\
&= \mathrm{KLUB}_{\mathbb{P}_s}(\mathbb{P}_t\|\mathbb{Q}_t^{s \to t}).
\end{aligned}$$

### B.4 TECHNIQUE 1

The approximation is made as follows:

$$\mathcal{D}_{\mathrm{KL}}(\mathbb{P}_{path}\|\mathbb{Q}_{path}^{T \to 0}) - \mathcal{D}_{\mathrm{KL}}(\mathbb{P}_{path}\|\mathbb{Q}_{path}^{T \to t \to 0}) = \mathbb{E}_{\mathbb{P}_{path}}\left[\log \frac{\mathbb{Q}_{path}^{T \to t \to 0}}{\mathbb{Q}_{path}^{T \to 0}}\right] \tag{29a}$$

$$\approx \mathbb{E}_{\mathbb{Q}_{path}^{forward}}\left[\log \frac{\mathbb{Q}_{path}^{T \to t \to 0}}{\mathbb{Q}_{path}^{T \to 0}}\right] \tag{29b}$$

$$\approx \mathbb{E}_{\mathbb{Q}_{path}^{T \to t \to 0}}\left[\log \frac{\mathbb{Q}_{path}^{T \to t \to 0}}{\mathbb{Q}_{path}^{T \to 0}}\right] \tag{29c}$$

$$= \mathcal{D}_{\mathrm{KL}}(\mathbb{Q}_{path}^{T \to t \to 0}\|\mathbb{Q}_{path}^{T \to 0}) \tag{29d}$$

Equation equation 29a assumes that $\mathbb{P}_{path} \approx \mathbb{Q}_{path}^{forward}$ where $\mathbb{Q}_{path}^{forward}$ refers to the distribution made by forward CTMC. In equation equation 29b, we assume that $\mathbb{Q}_{path} \approx \mathbb{Q}_{path}^{T \to t \to 0}$. It is important to note that we can use equation 29a as a formula for KLUB computation, as introduced in Algorithm 1. However, the results of JYS sampling schedule optimization show little difference between the two.

Compared to coarser sampling, KLUB computation can be organized as follows:

$$\text{KLUB}\left(\mathbb{Q}_0^{T \to t \to 0} \,\|\, \mathbb{Q}_0^{T \to 0}\right) = \mathbb{E}_{\mathbb{Q}_{\text{paths}}^{T \to t \to 0}}\left[\sum_{i \neq j}\sum_{u=0}^{T} H_u^{ij} \log \frac{R_u^{T \to t \to 0}(i,j)}{R_u^{T \to 0}(i,j)}\right] \tag{30a}$$

$$= \mathbb{E}_{\mathbb{Q}_{\text{paths}}^{T \to t \to 0}}\left[\sum_{i \neq j}\sum_{u=0}^{t} H_u^{ij} \log \frac{R_t(i,j)}{R_T(i,j)} + \sum_{i \neq j}\sum_{u=t}^{T} H_u^{ij}\log \frac{R_T(i,j)}{R_T(i,j)}\right] \tag{30b}$$

$$= \mathbb{E}_{\mathbb{Q}_{\text{paths}}^{T \to t \to 0}}\left[\sum_{i \neq j} \log \frac{R_t(i,j)}{R_T(i,j)} \sum_{u=0}^{t} H_u^{ij}.\right] \tag{30c}$$

In Eq. (30b), we utilized the fact that under $\tau$-leaping, $R_u^{T \to t \to 0}(i,j) = R_T(i,j)$ for $u \in [t,T]$ and $R_u^{T \to t \to 0}(i,j) = R_t(i,j)$ for $u \in [0,t]$. In Eq. (30c), the rate matrices are constant over intervals, allowing us to pull $\log \frac{R_t(i,j)}{R_T(i,j)}$ outside the summation.

### B.5 TECHNIQUE 2

Consider the meaning of $\mathbb{E}\left[\sum_{u=0}^{t} H_u^{ij}\right]$; it calculates the average probability of a transition from $i$ to $j$ occurring between time 0 and $t$. If we knew $\partial_u p(x_u = j, x_{u-} = i)$, this could be found by $\int_0^t \partial_u p(x_u = j, x_{u-} = i)\, du$. However, we do not have access to $\partial_u p(x_u = j, x_{u-} = i)$.

Fortunately, we do know the conditional transition rate $\partial_u p(x_u = j \mid x_{u-} = i) = R_t(i,j)$. Let's assume that there are maximally single transition of state in each dimension during the time interval, which is the assumption behind using $\tau$-leaping algorithm for DDMs (Campbell et al., 2022). Using this, we can rewrite Eq. (30c):

$$\mathbb{E}_{\mathbb{Q}_{\text{paths}}^{T \to t \to 0}}\left[\sum_{i \neq j} \log \frac{R_t(i,j)}{R_T(i,j)} \sum_{t < u \leq T} H_u^{ij}\right] = \mathbb{E}_{\mathbb{Q}_{\text{paths}}^{T \to t}}\left[\mathbb{E}_{\mathbb{Q}_{\text{paths}}^{t \to 0}}\left[\sum_{i \neq j} \log \frac{R_t(i,j)}{R_T(i,j)} \sum_{t < u \leq T} H_u^{ij}\Big| X_t = i\right]\right] \tag{31a}$$

$$\approx \mathbb{E}_{\mathbb{Q}_{\text{paths}}^{T \to t}}\left[\sum_{X_t \neq j} \log \frac{R_t(X_t,j)}{R_T(X_t,j)} \times R_t(X_t,j)\Delta t\right] \tag{31b}$$

Equation Eq. (31a) applies the Law of Total Expectation, and in Eq. (31b), we utilize the equation:

$$\mathbb{E}_{\mathbb{Q}_{\text{paths}}^{t \to 0}}\left[\sum_{u=0}^{t} H_u^{ij}\Big| X_t = i\right] \approx R_t(i,j)\Delta t,$$

where $\Delta t = t - 0 = t$. The approximation becomes exact when there are only one transition in each dimension during single interval, and this assumption is true for absorb transition matrix.

## C ALGORITHM

In this section, we present the main algorithm for Jump your steps (JYS).

### C.1 KLUB COMPUTATION

Please refer to Algorithm 1.

In the case of $k$-Gillespie, computing KLUB using Algorithm 1 requires the start timestep $s$, the end timestep $u$, and the breakdown timestep $t$. However, in $k$-Gillespie, the schedule is determined not by timesteps but by the number of generated tokens. Specifically, the start number of generated

tokens $k_s$, the end number of generated tokens $k_u$, and the breakdown number of generated tokens $k_t$ are provided as inputs. Once the corresponding timestep $t$ for a given number of generated tokens $k$ (i.e., $p(t|k)$) is determined, Algorithm 1 can be used to compute $\text{KLUB}(\mathbb{Q}^{k_s \to k_t \to k_s} \| \mathbb{Q}^{k_s \to k_u})$.

Note that the noise schedule determines the probability $p_t$ of each token being masked at time $t$, enabling us to compute the probability of $k$ tokens are unmasked at time $t$. Let $p_t$ denote the probability that a token will be unmasked at time $t$. Now, imagine that we implement the forward process $q_{s|0}$. First, we sample a value between 0 and 1 from a uniform distribution for each token; Second, tokens with values greater than $p_t$ are masked. The reverse process $p(t|k)$ can be derived similarly: assign a random value between 0 and 1 to each token and find the value of $t$ such that exactly $k$ tokens have values greater than $p_t$. This approach allows us to effectively sample $p(t|k)$.

---

**Algorithm 1: Computation of $\text{KLUB}(\mathbb{Q}^{s \to t \to u} \| \mathbb{Q}^{s \to u})$**

**Require:**
  $\theta$: Diffusion model parameters
  $s, t, u$: Timesteps, with $s > t > u$
  $p_{data}$: Data distribution
  $N$: Number of Monte Carlo samples
**Ensure:**
  KLUB: Computed KLUB value
1: Initialize $\text{KLUB}_u \leftarrow 0$ and $\text{KLUB}_d \leftarrow 0$
2: **for** iteration $= 1$ to $N$ **do**
3:   Sample $X_0 \sim p_{data}$                                    ▷ Sample from data distribution
4:   Sample $X_s \sim q_{s|0}(X_s \mid X_0)$                       ▷ Forward process to $s$
5:   Sample $X_t \sim p_{t|s}^\theta(X_t \mid X_s)$      ▷ If we use Eq. (29a), $X_t \sim q_{t|0}(X_t \mid X_0)$.
6:   Set $\Delta t \leftarrow s - t$
7:   Update $\text{KLUB}_u \leftarrow \text{KLUB}_u + \sum_j \Delta t R_t^\theta(X_t, j) \log \frac{R_t^\theta(X_t, j)}{R_T^\theta(X_s, j)}$       ▷ Eq. (15)
8:   Increment $\text{KLUB}_d \leftarrow \text{KLUB}_d + 1$
9: **end for**
10: Compute $\text{KLUB} \leftarrow \text{KLUB}_u / \text{KLUB}_d$                     ▷ Final KLUB value
11: **return** KLUB

---

### C.2 JUMP YOUR STEPS

Please refer to Algorithm 2.

### C.3 $k$-GILLESPIE ALGORITHM

To aid readers' understanding, we include the $k$-Gillespie algorithm (Zhao et al., 2024). Please refer to Algorithm C.3. The algorithm is structured as follows: first, the backward rate matrix is calculated. Then, the holding time for $k$ transitions is computed to update $t$, and $k$ transitions are performed sequentially.

## D EXPERIMENT DETAILS

**Golden Section**   The golden section search was stopped if the difference between the newly optimized $t$ and the previous $t$ was smaller than $T/2048$. The maximum number of iterations was set to 32, but usually, the iterations were completed within 8 steps.

**CountDown**   We use SEDD (Lou et al., 2024) for loss function and the DiT (Peebles & Xie, 2023) as a model architecture, the noise schedule followed the log-linear scheme proposed in the SEDD paper. KLUB computation was done with num_samples $= 2048$, and one golden section search took approximately 4 seconds.

**CIFAR10**   The pretrained model provided by CTMC (Campbell et al., 2022) was used. For CIFAR10, with num_samples $= 1024$, one golden section search took about 30 seconds.

---

Algorithm 2: Jump Your Steps

---

**Require:**
    $2^K$: Number of function evaluations
    $T, 0$: Maximum and minimum timesteps
**Ensure:**
    $K \geq 1$: Number of iterations
1: Initialize Timesteps $\leftarrow (T, 0)$
2: **for** $k = 1$ to $K$ **do**
3:     Initialize Timesteps$^* \leftarrow ()$
4:     **for** each pair $(s, u)$ in Timesteps$[: -1]$ and Timesteps$[1 :]$ **do**
5:         Compute $t \leftarrow$ GoldenSection $(t, \text{KLUB}(\mathbb{Q}^{s \rightarrow t \rightarrow u} \| \mathbb{Q}^{s \rightarrow u}))$
6:         Update Timesteps$^* \leftarrow$ Timesteps$^* + (t)$
7:     **end for**
8:     Initialize Timesteps$^{**} \leftarrow ()$
9:     **for** each pair $(t_i, t_j)$ in Timesteps$[: -1]$ and Timesteps$^*[1 :]$ **do**
10:        Update Timesteps$^{**} \leftarrow$ Timesteps$^{**} + (t_i, t_j)$
11:     **end for**
12:     Update Timesteps $\leftarrow$ Timesteps$^{**}$
13: **end for**
14: **return** Timesteps

---

---

Algorithm 3: $k$-Gillespie's Algorithm with Corrector Steps

---

**Require:**
    $\theta$: Diffusion model parameters
    $k$: number of token generated in a single step
    $L$: sequence length
1: Initialize time $t \leftarrow 1$
2: Initialize sample $x \leftarrow \text{MASK}^L$
3: **for** $i = 1$ to $L$ **do**
4:     Compute backward rate $r_i^l = \hat{R}_i^\theta(x, G_i^l(x))$
5:     Calculate total rate $r^l = \sum_{i \neq l} r_i^l$
6:     Sample holding time $\tau^l \sim \text{Exp}(r^l)$
7:     **for** $k = 1$ to $K$ **do**                     ▷ Make multiple state transitions
8:         Get dimension of transition $l^* = \text{SORTED}(\tau^l)[k]$
9:         Update state $x^{l^*} \leftarrow \text{Cat}(r^{l^*})$ where $r^{l^*} = \frac{1}{r^{l^*}}(r_1^{l^*}, \ldots, r_S^{l^*})$
10:     **end for**
11:     Update time $t \leftarrow t - \tau^{l^*}$
12:     **if** $t \leq t_{\min}$ **then**
13:         **break**
14:     **end if**
15: **end for**
16: Find most likely values for the ungenerated dimensions $x \leftarrow \arg\max_{x_0} p_t^\theta(x_0 \mid x)$
17: **return** $x$

---

**Monophonic Music**    The pretrained model provided by CTMC (Campbell et al., 2022) was used. KLUB computation was performed with 2048 samples, and one golden section search took about 20 seconds.

**Text**    The pretrained model provided by SEDD (Lou et al., 2024) was used. With num_samples = 256, one golden section search took 120 seconds.

# E   ADDITIONAL RESULTS

## E.1   EFFICIENCY OF THE JYS ALGORITHM IN SAMPLING SCHEDULE OPTIMIZATION

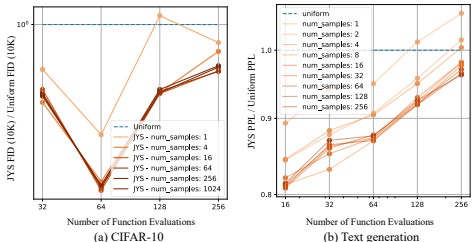 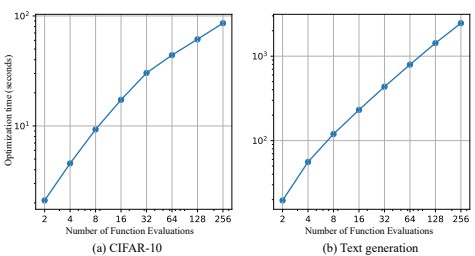

Figure 10: The effect of the number of Monte Carlo samples on the performance of the JYS schedule.

Figure 11: We measured the wall-clock time required for optimizing the JYS sampling schedule in a practical setup using a single 24GB NVIDIA RTX 3090 GPU.

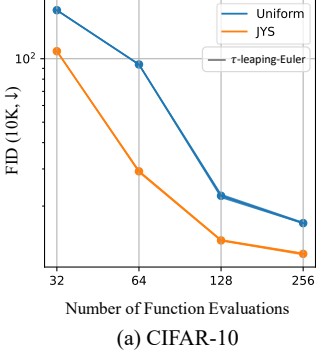 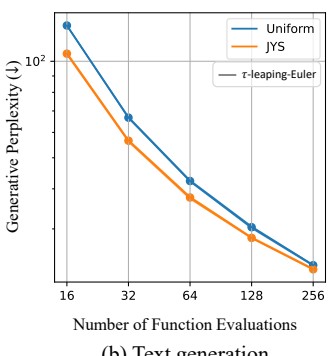

Figure 12: Error bars or standard deviations for the quantitative metrics are shown, with the maximum and minimum values represented as shaded regions and the mean value indicated by a solid line.

To inspect the practical utility of our algorithm, we investigate the optimization time to get JYS schedule. When optimizing the sampling schedule, there is a trade-off: increasing the number of Monte Carlo samples for KLUB estimation improves the reliability of the estimation but also increases computational demands. Our goal is to use the minimum number of samples necessary while maintaining reliable performance.

We conducted experiments on CIFAR-10 and a text generation task to evaluate this trade-off (see Figure 10). Our findings show that reducing the number of Monte Carlo samples to as few as 16 does not result in a significant performance drop. We hypothesize that the sampling schedule optimization is robust with fewer samples because the KLUB, which our optimization aims to minimize, does not vary greatly between the sample trajectories (see Figure 4).

Based on these results, we measured the time required for the JYS sampling schedule optimization on a practical setup using a single 24GB NVIDIA RTX 3090 GPU (Figure 11). Note that with more GPUs, the Monte Carlo sampling could be parallelized, further reducing the time. For a sampling schedule with NFE=64, the optimization took only 45 seconds in CIFAR-10 and 5 minutes on the text generation model (SEDD-small). In contrast, AYS required approximately 6 GPU hours for NFE=50 on CIFAR-10 and 32 GPU hours for ImageNet 256×256 with RTX6000 GPUs (Sabour et al., 2024).

Importantly, this time cost applies only to the initial sampling schedule optimization; once optimized, there is no additional inference cost.

### E.2 ERROR BARS AND STANDARD DEVIATIONS FOR QUANTITATIVE METRICS

In Figure 12, we report the minimum, maximum, and mean values from experiments conducted with three random seeds on CIFAR-10 and the text generation task. The results show minimal differences across the three seeds, demonstrating the robustness of our metrics to random seed variations. This

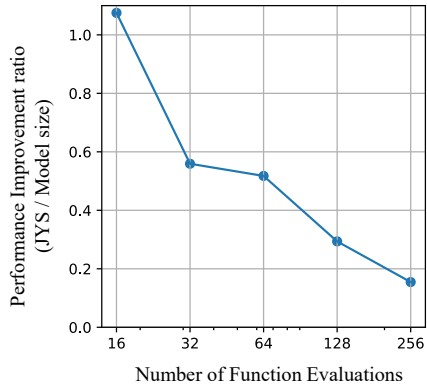
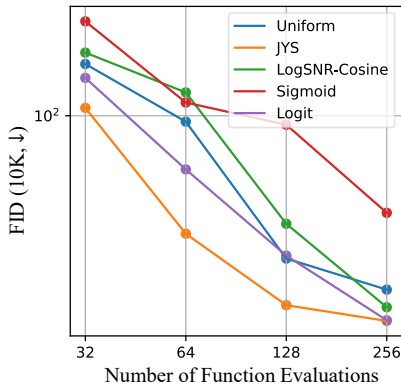

Figure 13: Relative performance gains from JYS compared to model size scaling.

Figure 14: Performance comparisons on CIFAR-10.

robustness likely stems from generating a sufficiently large number of samples (10K for FID and 1,024 for generative perplexity) before performing the measurements, thereby reducing variability. For CIFAR-10, the average difference between the maximum and minimum FID (10K) values is 0.37, while for text generation, the average difference in perplexity (PPL) values is 0.65—both negligibly small.

### E.3 PERFORMANCE IMPROVEMENTS ACHIEVED BY JYS

To demonstrate the impact of JYS, we compared its performance gains to those achieved by increasing the model size. In Figure 13, we show the reduction in generative perplexity (PPL) when transitioning from a uniform to a JYS sampling schedule for each NFE, alongside the reduction observed when scaling the model from SEDD-small to SEDD-medium. This comparison highlights the proportion of performance improvement attributable to JYS relative to that achieved by increasing the model size.

### E.4 COMPARISON WITH HEURISTIC SAMPLING SCHEDULES

To provide a more comprehensive evaluation, we conducted additional experiments comparing our approach with heuristic sampling schedules beyond the uniform baseline. Specifically, we tested a sampling schedule matching the signal-to-noise ratio (SNR) of LogSNR-Cosine and heuristic schedules inspired by the timestep sampling from (Esser et al., 2024), which allocate more sampling steps toward the middle (Sigmoid) and both ends (Logit). As shown in Figure 14, these heuristic schedules consistently underperformed compared to the JYS schedule, which achieved the best performance across all NFE values on both CIFAR-10 and text generation tasks. This underscores the challenges of improving sampling schedules solely through heuristic methods.

### E.5 JYS SAMPLING SCHEDULES FOR NFE $\neq 2^K$

In this subsection, we introduce a method for applying the JYS sampling schedule when the number of function evaluations (NFE) is not a power of two ($2^K$). Our proposed approach involves selecting an appropriate subset from an optimized sampling schedule. We will explain this with a concrete example for easier understanding. For instance, if we aim to perform sampling with NFE $= 48$, we first optimize the JYS schedule for NFE $= 64 = 2^6$. The next step is identifying a suitable subset from this schedule.

To select 48 steps from the NFE $= 64$ schedule, we proceed as follows: (1) all time points corresponding to NFE $= 32 = 2^5$ are included, and (2) among the additional time points introduced by JYS NFE $= 64$, we filter based on their KLUB values. Specifically, we add $t_{2i}$ in descending order determined by $\mathrm{KLUB}(\mathbb{Q}^{t_{2i-1} \to t_{2i} \to t_{2i+1}} \| \mathbb{Q}^{t_{2i-1} \to t_{2i+1}})$. This ensures to include $t$ such that most ef-

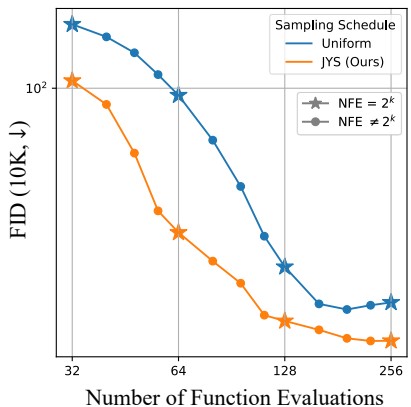

Figure 15: Quantitative performance comparisons on CIFAR-10.

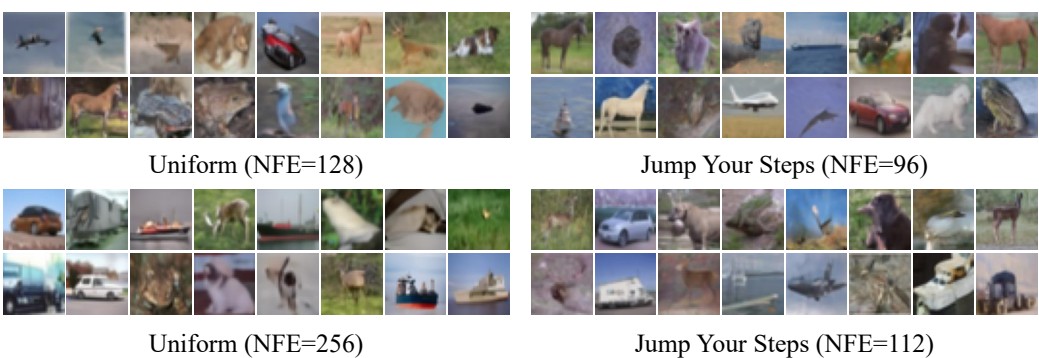

Uniform (NFE=128)  Jump Your Steps (NFE=96)

Uniform (NFE=256)  Jump Your Steps (NFE=112)

Figure 16: Qualitative performance comparisons on CIFAR-10. For the JYS schedule, the NFE was selected as the minimum NFE that achieves the same or lower FID compared to uniform.

fective in reducing our optimization target, the compounding decoding error (CDE), in the sampling schedule.

Figure 15 illustrates the results of CIFAR-10 image generation experiments using the JYS sampling schedule constructed by the above method. Notably, even when NFE is not a power of two, the JYS sampling schedule achieves comparable performance to the uniform baseline with only half the NFE. Figure 16 qualitatively compares the generation results using JYS sampling schedules with NFE equal to or lower than NFE = 128, 256. As can be seen from the figure, even when using a lower NFE, JYS produces generation results of the same quality level.

### E.6 QUALITATIVE RESULTS

In this subsection, we present qualitative comparison between Jump Your Steps and Uniform sampling schedule under various NFEs.

**Toy 2D datasets** Figures 17, 18, and 19 display generated samples under various NFE and sampling schedules. Each models are trained with the uniform transition kernel. All results were generated using the Euler $\tau$-leaping sampler. It is evident that JYS produces higher-quality samples compared to the uniform sampling schedule.

**CIFAR10** Figure 20 shows generated images with various NFE and sampling schedule. All results are generated using Euler $\tau$-leaping sampler.

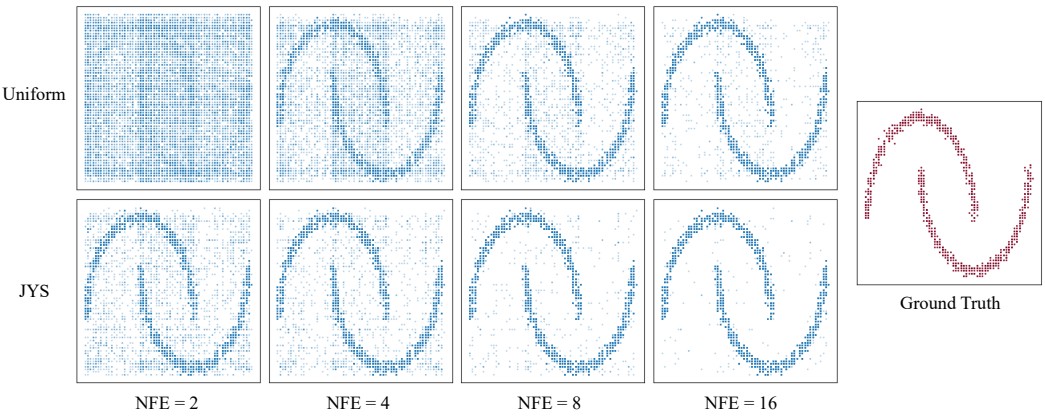

Figure 17: **Toy 2D dataset results - Moons.**

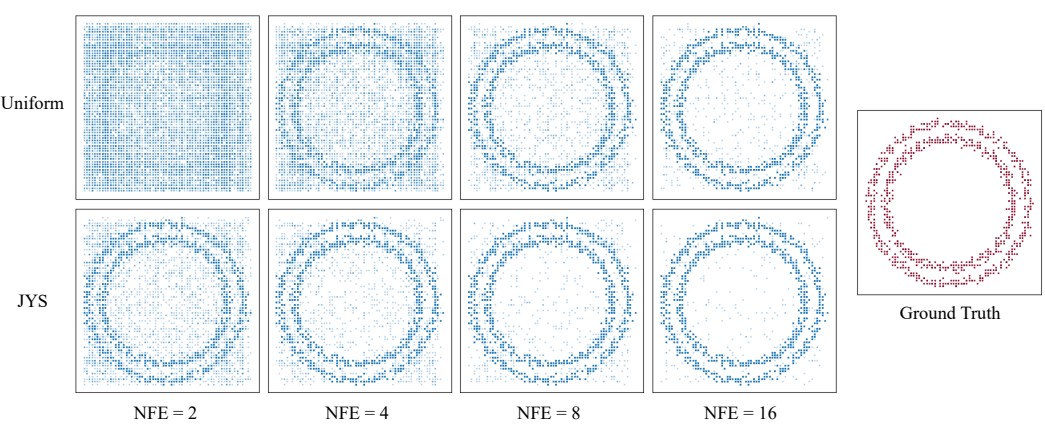

Figure 18: **Toy 2D dataset results - Circles.**

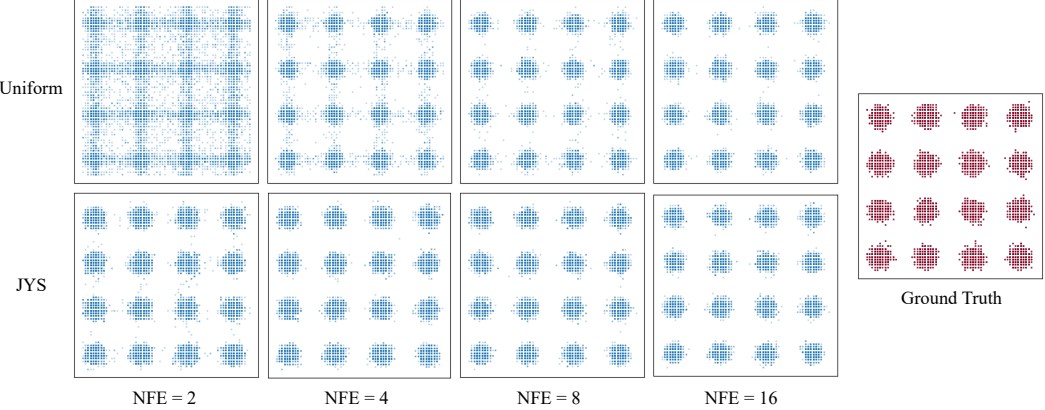

Figure 19: **Toy 2D dataset results - Gaussian Mixture.**

**Text** Figure 21, 22, 23, 24 show generated text samples with various NFE and sampling schedule. All results are generated using Euler $\tau$-leaping sampler.

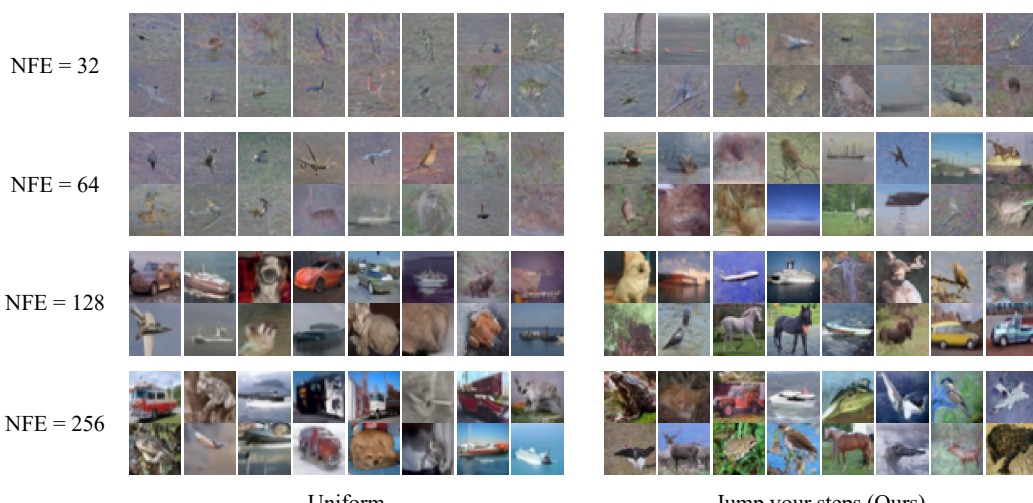

Figure 20: **CIFAR10 results.**

', while playing for Vevian, a Habs minor when he was sent to to Providence, Nova Scotia.\n\nThere were signs, while seated in the stands, of a self awareness between Giiroz and Vevian, who were son who raised him how the three could possibly stand in front of Habs's jersey to help ease his mind.\n\n"Isn't an insular thing, " Giiroz said. "They're your teammates, and they love helping you."\n\nAdvertisement\n\nJosh willre B. Rogers mentioned Dustin Quick\'s name up on Bruins jerseys. The team plays Wednesday night, Feb. 4, in Bel Air N.J. J. M. Stringer/AP\n\n"I don't think anybody'd be mad at anybody to resell them like that," Giiroz said. "Of things."\n\nThe aren't talking of Team No. 3. Currently the team\'s name is A6, the Gi five nickname, but already there's been enough sales to name more than 100 players, in addition to using the team\'s name in games.\n\n"It's really important and I've got a lot of people who don't like us at all, but some people may not and all I am saying is that we are selling them our jerseys, these jerseys can look different, there's a lot more trust than people would like it to have, " Giiroz said.\n\n"I don't speak for the guys that even have them, but the will doesn't have to feel for them as the ego does," Granoff said."Just don't ever concern or complain."\n\nAdvertisement\n\n"I was walking around town, they can't be named U27 and A34 and what could I say? Two years ago they mentioned about them that they were the Bruins No. 3," he told NewHockey's Plex Myers.\n\nIt appeared first on Jarusthir\'s jersey when his car, recently registered seven times, was stolen in 2012.\n\nThe forwardian Johnson said he\'s one of the six in the No. 3 team, "because I can't get my name. I am one of the game. It was sold and the decision they made is theirs."\n\nHealth is number one during pregnancy in America, with a lack of complications, such as heart disease, heart problems and health or mental problems. Finally, the Health Institutes is slowly diluting itself in its own way: Last year, the health care debate in the country was challenging yet electrified Liberal leaders. (Still, anyone who doubts that public health care will be at hand could easily claim that it fails to deliver no matter how hard you work.)\n\nThe big guns on health care when it comes to health America's health crisis have been the dismantling of some traditional government programs.\n\nThe Health and Human Health Service is drawing up a starting line here in Washington this week for a 180-page health reform proposal. The White House is still drafting its latest draft law, which will have its first version by the end of the year.\n\nThe economist is in charge of the department of health care and troubled Medicaid Services, which plans to eliminate up to 750,000 employees next year — accusing health insurance buyers of taking profits away from communities. Mr. Price, a former GOP congressman in the House, said this interview that he would work on expanding exclusive hospital care for sick patients, citing concerns over either public or professional spending on drug use.\n\nMs. A. also contends that there is a surge in medical care and highlights challenges such as rising insurance premiums, growing market sizes and shrinking health benefit programs in key markets such as hospitals, universities, schools and hospitals.\n\n(Fix: Watch the full video to calls out on uninsured's rise on gov)\n\nIn an e-mail, Mrs. C Perry countered that any new overhaul of pay would typically "seriously affect our workforce and its cost and administrative efficiency."\n\nThat\'s because strong health care programs have bolstered smart young people as an alternative to older adults — thanks, in part, to pressures such by Democrats and the Congress to share savings on top-stealing causes.\n\nFirst, first and foremost, the health, today's economy — health care, which is being replicated heavily, sees a rise of information technology workers who are fluent in certain kinds of work — peaked in 1996. That number of health employees reached 1.2 million in 2010. The current number is smaller than most of the working-age workforce, by registered age.\n\nAmerica\'s young people\n\nAlso, the Federal government restricts healthcare delivery to certain low-income people so that they can not be forced into the insurance market. Two-thirds'

Figure 21: Generated samples using uniform schedule (NFE = 64).

' organization used it for a marketing campaign, some of them. The office was sent to to Providence, arcana. Overall, the organization generated $12 million in revenue for 2015 and $8 million in tax revenue spending.\n\nGreg Cohn, who was one who first knew how the organization came to Boston, came back in Toronto and now visits Boston to help players.\n\n"That man hates me," he said. "This organization, the player, they come from special circumstances and often in the world, and they love helping you."\n\nHe assured us they will be found. But when I heard their name, I had to wonder if they would work to build the organization, and bring those guys out of the shadows themselves and bring a little Pete Noles to the Boston scene.\n\n"I don't know if we are containing the cream, but that was exactly what I thought when asked to speak when that organization went to Toronto," said Eric Krodyns, the team\'s director. AIM then asked Cohn why he was still in Boston. He declined to comment on the name and structure of that office.\n\nOnce the team began the reporters to Boston,, the staff had problems raising the idea of calling into those reporters, he explained. He said he chuckled after listening to many plays at Soarin's. "But I am the man that we are identifying with."\n\nOne can completely swear he wouldn't. The Royals are destined for nothing like the White Sox, and they are be heading back to The kid's dreams only if teams are willing to bless that and turn them into something that will allow them to totally get there. Wow, not easy, does it?\n\n"Just don't ever let the scouts use their names on the field, don't let them leave town, let their not show in a lot of games. A candid scout is what I've heard a million times, but that's a very serious thing," he told New America Baseball's Scott Davis. "[It is a] catalyst for either, for a split second, a six or seven people that just might be there. That organization has given us a chance to get back into the rotation in the spring. Our team has showed us a really, really great game."\n\nAs of how to live out this season of the Mets? This is pretty chime. The real road is to make the World Series. They really surprise other teams as always. Sources from their three league report told Yahoo Sports last month the Mets' schedule is now second in its baseball calendar. Last year they're not in the 11th, yet using baseball calendar.\n\nStill, anyone that should be awarded a win Sunday night at home could easily not make it over weekend, no matter how hard they work.\n\nBut Larry Anthony, CEO of Major League Baseball Baseball of America said it's hard not to understand why some teams could not easily win during the regular season.\n\n"We should stop losing, here in Toronto, three days a week. That can\'t be in for half."\n\nGet the latest in USA our Page View Map for first looks at the European and American leagues.\n\nLead image RICA / taken under a Creative Commons.<|endoftext|>Toronto's craft sector will gross up about 2.92 percent this year as foreign-owned buyers are taking starts away from U.S. regional chains, a Toronto bank in a new study said Friday.\n\nThere compounding is an undersupply of new restaurants, as new restaurants next expect to take hold next year. By then, Service Canada already is set to double as a U.S.-based market — now the fastest-topping craft market — and about 29 restaurants in Canadian craft markets will be added this year, going by 6.48 percent to 14.51 percent, CIBC economist with HBC's CIBC Corp. said.\n\nThe eOBA study led by CNAB Brendon Barley said new restaurants typically "seriously drain our workforce and demoralise the restaurant industry."\n\nThat\'s last fall\'s report from 2009 to March when more than 700 restaurants are currently insupplied, in part because of pressures such as an internet industry struggling to place value on top-staple craft.\n\n"First and foremost, based to Statistics Canada's data — which has tracked which industries are investing heavily, including the rise of cheap Canadian workers, new Canadians, major property market reforms — there are just over 150 craft restaurants across the continental U.S. in 2010. The craft sector is Canadian in most descriptive terms," Barley noted.\n\n"That\'s where people are most prepared to work and businesses are being forced to try to adapt themselves," he said.\n\nOne of the other ramifications is a poor Canadian pace'

Figure 22: Generated samples using Jump Your Steps schedule (NFE = 64).

'. I still like it as though. But now\'s the right time for me to say yes. And the right time for to say yes, the right time to say yes is to have a good time for me to finish this question."\n\nDonald said, Trump emerged from meeting his father in a private school, took parting shots at one of the few people who he has a connection with. He\'s so strong, and he\'s one of the strongest people he\'s seen.\n\nIt\'s like Donald Trump is over it\n\n"It\'s been a 100 percent experience," Trump told CNN after he shot him, "and it\'s not all, I think, it\'s just me. But I feel like I made me feel like something. I made me feel like something. But I\'d never felt this way, I\'d never ever need to go. But almost fading away, right now he\'s feeling something special."\n\nThis is my father\n\nTrump calls Trump one of the heen talks he\'s ever seen. "Then he did some words like, and he asked me a question, "and then he said, "I don\'t need to you, just shoot me." I said, I\'ve talked to at least one man, I can\'t wait to do that." Trump\n\nAre we going to want to be his father to play on?\n\nTypically in interviews, Trump says you have to ask, and they won\'t change anything. But he says he does want to be his father in the world. Even if he\'s depressed by what\'s happening, he says there is more from him. He added, "I really want to get myself out there and be the best person in the world, so go out there."\n\nBut is what want? Trump, says, "I don\'t have to worry much about myself anymore, but I just want to make my day better."\n\nEveryone\'s going to hate him, Donald Trump\n\n"I am a great moment when people say I hate," he said. "I also know him, the world has a saying called, \'Here\'s my son,\' and I\'m gonna like it. But also, I\'m really really proud of what he does. He brings so much. His life to me that makes him me. A hero to the world. But all the conversation going through, it\'s almost like I find out who his father really is. But then I go just don\'t he. Just have that son, and be a different dad. That is going to happen eventually.\n\nTrump says change, "sometimes it just takes time, and then sometimes it does, but I think he\'ll be different, and he will be the new leader. I hope he will be my mentor. And eventually, I may just be his replacement," Trump added.\n\nHe\'s going to have a Republican convention here, anyway\n\nNobody will tell you how big an impact that Donald Trump made in Cincinnati because he had because of the first Republican convention in America. "I said to all of the white people, much of all of the white people, I was like, \'You don\'t know enough about these people.\' I, to mention all that, tried to sound like a little too obvious and too politician," he said on CNN.\n\nTrump denies that kind of thing. "You should come across it."<|endoftext|>The Donald Trump will meet in January on Russia\n\nPresident Donald Trump leaves the House on Friday January 20, 2017.\n\nWASHINGTON, DC (WASHINGTON, Jan 20, 2017) —He has never been a man, but a partisan, both ways of looking on the scale of controversies in 2016 and 2016.\n\nWhile Mr. Trump touted his vision on such issues, he retired from last year's presidential election, even after he was earlier this year fired by members of the FBI and James Comey sent a letter in which he recommended Clinton not a presidential nominee.\n\nThen last month, Trump denounced the timing of his letter, say it was an obstruction only as it was not newsworthy.\n\nEven as he met with the FBI, Donald Trump rejected the apology to the media. (Reuters)\n\nMr. Comey, the man who led the Russia probe, said on Meet With The Press that he met with frontrunner, Hillary Clinton 2016, because it was the best way to make concessions.\n\n"I thought she would just something to me because she saw the favoritism of me, and she did so little for Clinton. That's not the deal," Trump said of him last week.\n\nMeanwhile, Mr. Comey is not a Democrat. "You know, this is going to be a great transition," Mr Mr. Comey's statement said, and it informed the president that he will next meet on Russia.\n\n"It's winding up a very long time, very long," Mr. Comey said. "And'

Figure 23: Generated samples using uniform schedule (NFE = 256).

'Story highlights Bush and Clinton as though they have more things to do\n\nBut Obama finally leads\n\nSome 47 percent of voters say they split the popular vote in November if the United States president wins the White House\n\nAccording to the latest Fox News poll, Americans everywhere from about 28 to 55 believe they would know if the other 48% of the U.S. has been president\n\nIt\'s clear why voters, on the one hand, believe that it\'s the most difficult years, why they believe Obama is president.\n\nObama has not spoken for as many months and says he cares deeply about him. More recently, he says months turned to me to nothing.\n\n"You\'ve got to wait until or until next month knows what\'s all going on?" he says.\n\n"I\'m staying here in 14 more years," Obama replies. "I don\'t think right now that\'s the way to go."\n\nBack to 2008\n\nBy the vast majority of the vote, Obama\'s up 44 percent. His support in the U.S. is about 16.2 percent, less than three points recorded by the Presidents George W. Bush and , though his leads has already climbed to about 50 points. That\'s one of those numbers that was crucial to Obama in Virginia.\n\nIf Obama is now leading Clinton in an effort to capitalize on the last presidential victory in November, Democrats won\'t have a victory, and they won\'t have anything official at all.\n\nThere are four notable candidates in the White House, and let\'s just ask what four candidates might be. Former President. W. Bush? We really know what he is: He said Obama was one of the candidates in the debate. The candidates were named.\n\nBut former President. W. Bush, "Especially with presidential politics, doing this is impossible." Former U.S. Gen. George E. Bush Obama maintained his clear lead in this debate. (Getty)\n\n"I think we have probably just come close to winning," he said. "I also know that many the people have a hand-to-hand and some of the number of other candidates is rising. But also, I guess I don\'t know what Bush would think of me, and I don\'t know what that would say, and we have to try on. But all the issues going through a president that I think I will overcome are really, really tough. But then I still just don\'t think anyone really knows who he is."\n\nCritics of the status quo question the Obama presidency. How the circumstance will change: "Actually, I think Washington will change the way it might get but I think people are suffering from poverty and lot of problems and I don\'t think this country will be so successful. I think other countries may not be taking care," said Dale Schleulich, past president of the Republican factions here, Texas and Wisconsin. (AP)\n\nThe southern states are key elections in a broader political strategy because they are not part of North America.\n\nPresident U.S. Obama won the support of several of the eastern states, including states in Ohio and Pennsylvania.\n\nSack has been raking in inroads in California, although he recently secured a majority in Wisconsin.\n\nOregon also is outshot in the U.S. presidential primary.\n\nThe political battle across America is tight. (The Texas debate is still on.)\n\nThe Ohio showdown: Ohio\n\nThere are polls indicating that Obama believes Republicans won\'t lose again four times next year, with maybe none voting the other way. However, he might take in a majority of the western states, leading up to Ohio in 2016.\n\nWhere the mirror may look\n\nSixty-six percent of Democrats say they\'re leaning in this election, according to Gallup.\n\nDemocrats have meanwhile been consistent on trade and national policy, incorporating racially polarized Republican policies.\n\nIn Vietnam this fall.\n\nJapan and South Korea benefit from a new U.S. relationship with Asian rivals, as China explores its role.\n\nJapan qualifies as trade ties with China, where there\'s rising the ante to the conflict. (Reuters)\n\nN.J. Gov. Chris Christie takes over the federal government this June. The state is in for a tough task as president-elect, and political experts expect he\'ll make an upset bid if Republicans win.\n\nVoters across the U.S. overwhelmingly favor him- for president and liberal candidate Bernie Sanders of New York, won by a landslide. Sanders has secured nearly 60% of Democratic primary voters.\n\nPresident George W. Bush isn\'t a good Cabinet secretary either, even further this year though Clinton has a great lead in the U.S. Senate -- which made him the party\'s leader in the race. A Gallup poll of the 2012 race earlier this year showed he is struggling to keep up with his rivals.\n\nGenerally, a mere 47% say the least say in some'

Figure 24: Generated samples using Jump Your Steps schedule (NFE = 256).

