# OpenReview forum: "Jump Your Steps: Optimizing Sampling Schedule of Discrete Diffusion Models"
_ICLR.cc/2025/Conference — ICLR 2025 Poster_

### Official Review · Reviewer_VTwh · 2024-11-03

**Soundness:** 3
**Presentation:** 3
**Contribution:** 2
**Rating:** 8
**Confidence:** 4

**Summary:**

The paper describes a way to optimize sampling for discrete diffusion models. The idea is to select a good subset of optimal sample times based on an optimization criterion. Some theory and some experiments are given.

**Strengths:**

The direction of this research is new and interesting.

**Weaknesses:**

Some of the theory parts confounds heuristic and rigorous justification (by ignoring some long-time dependencies without actually emphasizing that this is a working simplification). Thus, some assumptions and/or limitations are not highlighted, which is making the paper look weaker. ----[update] this was addressed in the discussion, with reference to appendices

Also, I am not completely convinced with the interpretation of some of the experiments, but this may be just me. --- [update] the authors clarified and promised to amend some figure

**Questions:**

1) line 189 saying "motivated by this observation" is not correct/precise I think, given that you list a bunch of factors at work and you just focus on one? the only motivating point is the dependence on sampling methods, and not the others.

2) line 202 "if we ignore the accumulated error from the previous steps": why should we ignore it? can you formulate this as an assumption somewhere?

I think that the paper's theory part might have very non-sharp estimates in case this error can't be ignored, can you convince me otherwise?

3) can you comment on when is the inequality in (7) sharp? I think this has to do with question 2.

4) line 216-217 "this theorem suggests" -- I think that it'd be accurate to say "based on the case that inequality (7) is sharp, we are tempted to hypothesize" and to mention that in case accumulated error from previous steps is negligible, then this heuristic becomes stronger.

Note that in applications in which the gap in (7) is large, then theorem 3.1 gives a lax and uninformative upper bound and this heuristic has no reason for holding.

5) about Theorem 3.2, when is the inequality in (8) sharp? Upper bounds like KLUB are useful to the extent to which they are close to sharp. So this limits the range of applicability of (8), and therefore it would be good to have a description of cases in which this is sharp.

6) about the hierarchical time breakdown (described in paragraph 3.4 and summarized in Figure 3) : this is one possible time breakdown strategy. Can you comment on alternative strategies, or on how this one is better than, or equivalent to, other strategies?

7) line 298-299 what do you mean by "computational complexities"? This is vague and easy to misinterpret for a reader. I think you mean the thing mentioned in lines 350-360? If so, I suggest moving those lines up here, so that the reader has a concrete thing in mind since the beginning.

8) about the formula/equation at lines 308-309: what does the "approximately equal" sign actually mean? and why or under what hypotheses is this approximation valid? You mention something in the preceding paragraph, but this is not giving a clear cut answer to what approximation you have in mind, and on what are the underlying assumptions behind it.

9) about figure 9, I am puzzled/perplexed by the cases (a) and (c):
In (a), why do you think that the JYS schedule has so dense subdivision in the first interval? I don't see anything in the "ground truth steps" graph that would indicate that this is optimal.
In (c), why is the JYS deciding to just not include any timesteps in the first interval? In there, I think that the Ground Truth Steps part has a very dense time dependence, at least that's what I see in the figure.. so how do you justify this discrepancy?

To summarize, both points (a) and (c) show that JYS has high discrepancy in its "optimized" timestep choice, compared to "ground truth samples", so it would be good to understand what this means, how do we interpret this.

---

> ### Author Response · Authors · 2024-11-22
>
> Thank you for recognizing our work as new and interesting.
>
> > **[W1]** Some of the theory parts confounds heuristic and rigorous justification
>
> We intentionally structured our paper to separate these elements: Sections 3.1–3.3 present the theory supported by rigorous justification, while Sections 3.4–3.5 introduce practical simplifications to make KLUB optimization tractable. To clarify this distinction, we explicitly stated it in the introduction to Section 3 in the revised manuscript.
>
> > **[Q1]** line 189 saying "motivated by this observation" is not correct/precise
>
> Thank you for pointing out this potential source of confusion. To address this, we simplified line 189 to avoid ambiguity: "In general, CDE depends on the timesteps, so we hypothesize that optimizing the sampling schedule can reduce CDE during the generation process."
>
> While it is true that CDE also depends on factors like the data distribution and corruption kernel, these are determined during the pretraining phase and fall outside the scope of our work. Including factors unrelated to our study's objectives would only introduce unnecessary complexity and potential confusion, so we decided to remove this redundant information.
>
> > **[Q2]** (line 202) "if we ignore the accumulated error from the previous steps": why should we ignore it? can you formulate this as an assumption somewhere?
>
> Thank you for this thoughtful question, which helps clarify the goals of our work. The statement in L202 is not an assumption but rather reflects our primary focus. The "accumulated error from the previous steps that affects consecutive steps" occurs when a generated $x_t$ does not belong to the distribution $q(x_t)$, meaning there is a mismatch between the training dataset and the generated dataset. We refer to this error as *exposure bias* [1].
>
> While exposure bias is indeed an interesting problem to address in generative model, our work focuses specifically on resolving the CDE caused by parallel decoding. Therefore, we chose to set aside exposure bias in this study. To enhance clarity, we included a discussion of this point in Appendix A.2 of the revised paper.
>
> > **[Q3, Q4, Q5]** (Tightness of the upper-bound, Theorem 3.1 and 3.2) I think that the paper's theory part might have very non-sharp estimates of error, which gives a lax and uninformative upper bound.
>
> Thank you for raising this insightful question. While we understand concerns about the sharpness of bounds, we respectfully disagree that non-tight bounds are inherently uninformative. Even if the KLUB derived in Theorems 3.1 and 3.2 is not always tight, it still serves as a **highly useful upper bound**.
>
> A notable example of a similarly non-tight but useful bound is the ELBO. The ELBO only becomes tight when $D_{KL}(\mathbb{P}||\mathbb{Q}) = 0$. Despite often being far from equality—particularly during initialization—the ELBO remains a cornerstone for optimization in many generative modeling tasks.
>
> Similarly, KLUB may not always be a tight bound, but it is still useful for sampling schedule optimization. The equality conditions for the Theorem 3.1 and 3.2 are as follows (Appendix B.1 and B.2):
> - Theorem 3.1 holds when, for any $i \in \{1, \cdots, N\}$, $D_{KL}(P_{t_{i-1}|t_{i}}||Q_{t_{i-1}|t_{i}}) = 0$.
> - Theorem 3.2 holds when, for any $t \in (0, T]$, $D_{KL}(P_{t-dt|t}||Q_{t-dt|t}) = 0$.
> - It is worth noting that, as shown in Appendix B.1.1, the bound in Theorem 3.1 is **tighter** than the negative ELBO bound commonly used in training diffusion models with fixed timesteps.
>
> Since $\mathbb{Q}$ represents the distribution generated through parallel decoding, it can never be identical to $\mathbb{P}$, making the inequalities generally non-tight. Despite this, we empirically demonstrated across various scenarios that optimizing with these bounds leads to performance improvements. This strongly suggests that, like the ELBO, our bounds serve as meaningful and effective optimization metrics.
>
> > **[Q6]** About the hierarchical time breakdown (described in paragraph 3.4 and summarized in Figure 3) : this is one possible time breakdown strategy. Can you comment on alternative strategies, or on how this one is better than, or equivalent to, other strategies?
>
> As part of our preliminary experiments, we explored an alternative strategy—hierarchical merging—where smaller, fragmented sampling schedules are combined into a coarse sampling schedule. However, our experimental results indicated that this approach often gets stuck in local optima and fails to deliver significant performance improvements. We will include this discussion in the revised version of the paper.
>
> Even if hierarchical breakdown strategy not a guaranteed algorithm for reaching the global optimum, it is a heuristic algorithm designed to perform optimization efficiently; Through extensive experiments, we demonstrated that the hierarchical breakdown strategy is an effective approach for reducing CDE and improving sampling quality.

---

> ### Author Response · Authors · 2024-11-22
>
> > **[Q7]** (line 298-299, lines 350-360) What do you mean by "computational complexities"? This is vague and easy to misinterpret for a reader. I think you mean the thing mentioned in lines 350-360? If so, I suggest moving those lines up here.
>
> Thank you for suggesting ways to improve the clarity and flow of our paper. To address your concern, we have clarified the computational complexities at the beginning of Section 3.5, where we introduce Techniques 1 and 2. Specifically, we now explicitly state that these techniques aim to reduce computational complexity by addressing two key challenges: (1) the need for multiple Monte Carlo samples to achieve reliable KLUB estimation, and (2) the high computational cost of each sampling step. This provides readers with clearer context from the start.
>
> > **[Q8]** (lines 308-309) What does the "approximately equal" sign actually mean?
>
> We sincerely apologize for the oversight. Unfortunately, we mistakenly omitted the section on the approximation from the appendix. Thank you for giving us the opportunity to correct this mistake. We have added a detailed explanation in the appendix B.4. The key assumption behind the approximation is:
>
> $P_{paths} \approx Q^{forward}_{paths}\approx Q^{T \rightarrow t \rightarrow 0}_{paths}$
>
> As we referred to this as a *technique*, the approximation is intended to make the KLUB optimization feasible rather than being rigorous or precise. The second approximation, in particular, is quite strong. However, even without relying on this approximation, it is still possible to perform the sampling schedule optimization, by using $Q^{forward}_{paths}$ when sampling $x_s$, and $x_t$ from algorithm 2. However, we found that this approach does not result in a significant performance improvement compared to simply using the KL divergence. As such, we opted to use the KL divergence for simplicity. A detailed discussion of this can be found in the appendix B.4.
>
> > **[Q9]** Regarding Figure 9, how should we interpret it? What does it mean? Why the JYS sampling trajectory is not aligned with ground truth sample?
>
> First, we would like to clarify that the intent of the referenced figure was not to demonstrate similarity between the ground truth sample and JYS sampling. In fact, there is no inherent reason for the two to necessarily align. For example, in Figure 9(c), the early stages of ground truth sample trajectory are highly random and involve frequent transitions. However, because the conditional mutual information in this range is low, the CDE is small, making it a region where large steps are feasible. This is why JYS exhibits large jumps in the early stages, which accounts for the significant differences observed in (c).
>
> The intention of this figure was to intuitively illustrate the characteristics of the transition kernel through the ground truth sample trajectory and to present the optimized JYS schedule. It was provided as a visual aid to help explain the JYS schedule interpretation based on conditional mutual information mentioned in the main text. However, we realize that showing the ground truth sample trajectory might give the incorrect impression that they should align. To avoid confusion, we will revise the figure by removing the ground truth trajectory and retaining only the JYS schedule.

---

> ### Comment · Reviewer_VTwh · 2024-11-25
> **further discussion**
>
> Thanks, this was thorough and I am satisfied with the answers to most points, here are the ones left a bit unsatisfactory by your reply:
>
> [Q6]: I understand that you tried different ideas, but can you go further and say "the one we selected is 'best' in the following sense [...]" ? It'd be great if one had a provable result (a theorem) stating hypotheses or settings in which one can't do much better than hierarchical time breakdown.
>
> [Q7] Please in the final version refer to 3.5 when you talk about the "complexities", for the benefit of the reader!
>
> ---------
> I think that the explanations from the appendices and your stated amendments are rather convincing, and I'll be willing to defend the publication of this paper to a higher level than before this interaction. To highlight this 'numerically' I'll raise my score to "8: accept, good paper"

---

> ### Author Response · Authors · 2024-11-25
>
> We are so grateful to hear that most of your concerns have been resolved. Thanks to your detailed feedback, our paper has become much more solid than the paper that we submitted. We truly appreciate your time and effort. Thank you so much!
>
> Below are our responses to your questions:
>
> > **[Q6]:** I understand that you tried different ideas, but can you go further and say "the one we selected is 'best' in the following sense [...]"? It'd be great if one had a provable result (a theorem) stating hypotheses or settings in which one can't do much better than hierarchical time breakdown.
>
> For the final version, we will make an effort to construct as much theoretical reasoning as possible for why the hierarchical breakdown strategy is preferable. However, we’d like to clarify that the aim of this paper is not to claim that hierarchical breakdown is theoretically the best optimization strategy. Rather, there are specific cases where the breakdown approach is more beneficial. We believe that even identifying these special cases can provide more intuition about why this strategy works.
>
> > **[Q7]:** Please in the final version refer to 3.5 when you talk about the "complexities," for the benefit of the reader!
>
> In the final version, we'll make sure to refer to Section 3.5 in the parts where "computation complexity" is mentioned, so that readers can clearly understand what is meant by complexity.

---

> > ### Comment · Reviewer_VTwh · 2024-11-25
> >
> > This sounds reasonable, thank you for taking into account my suggestions!

---

### Official Review · Reviewer_ibjS · 2024-11-03

**Soundness:** 4
**Presentation:** 3
**Contribution:** 3
**Rating:** 8
**Confidence:** 4

**Summary:**

This paper introduces Jump Your Steps (JYS), a method for discrete diffusion models (DDMs) that enhances generation quality by finding optimal sampling schedules when using efficient parallel sampling methods. Parallel sampling methods such as τ-leaping can speed up DDM inference, but they introduce a Compounding Decoding Error (CDE) that demotes sample quality. The authors address this by deriving a practical upper bound on CDE and developing an efficient schedule optimization algorithm that minimizes this bound without additional computational overhead. They perform experiments on image, music, and text generation tasks to show that the JYS approach improves sampling quality across different transition kernels using fast sampling methods like τ-leaping or k-Gillespie.

**Strengths:**

The main idea of this paper is novel: the authors suggest reframing DDM sampling schedule optimization as a CDE minimization problem, introducing a Kullback-Leibler divergence upper bound (KLUB) metric to make the optimization tractable for improving sample quality in fast DDM sampling methods. The theoretical development seems sound, establishing connections between CDE minimization and path-space probability measures. This work is particularly interesting because it requires no architectural modifications or additional inference costs, unlike existing methods. As for the empirical validation, the paper includes an evaluation across diverse domains showing consistent improvements in the sample quality when using the same number of function evaluations (NFEs) as a baseline approach.

**Weaknesses:**

1. The empirical analysis can be improved:
 - While improvements in the generation quality in terms of FID or perplexity scores are shown, comparisons against other quality-improvement methods (e.g., predictor-corrector approaches mentioned in Section 1), and an explicit study of their computational costs are missing. The paper should either include such comparisons or justify why they aren't necessary.
 - The trade-off between schedule optimization time (or its memory requirements/end-to-end generation time) and sample quality gains isn't studied, making it difficult to assess the method's practical utility.
- No error bars or standard deviations are provided for the quantitative metrics. This is important since generative models can have high variance in their outputs and metrics. An elaboration of the significance of the reported results would be appreciated.
-- The comparison is limited to a uniform scheduling baseline. Would it be possible to compare against an adaptive step size integrator method (Runge–Kutta method modifications if they are applicable to the discrete case) or other works like a recent one [5]? It’d be great to include this or explain why such comparisons aren't applicable or necessary.
2. The paper doesn't clarify how the diffusion model formulation (Eq. 1) relates to other DDM variants from [2, 3, 4]. Such discussion would be appreciated.
3. The single-state transition assumption, while mathematically convenient, limits the method's applicability: it can lead to suboptimal schedules for fast-changing processes. It’d be great to see ideas on possible extensions to handle multiple transitions while maintaining computational tractability.
4. Some technical details are difficult to follow or verify. For example, the authors introduce two techniques for KLUB computation in Section 3.5 but don't explain when each is preferable. It looks like both techniques are combined in Algorithm 1. No discussion of whether the hierarchical breakdown strategy converges to globally optimal schedules. The clarity of the paper could also be improved. There are suggestions on this in the Questions section.
5. No code is provided for reproducibility.


[1] Hoogeboom, Emiel, et al. "Argmax flows and multinomial diffusion: Learning categorical distributions." Advances in Neural Information Processing Systems 34 (2021): 12454-12465.

[2] Santos, Javier E., et al. "Blackout diffusion: generative diffusion models in discrete-state spaces." International Conference on Machine Learning. PMLR, 2023.

[3] Austin, Jacob, et al. "Structured denoising diffusion models in discrete state-spaces." Advances in Neural Information Processing Systems 34 (2021): 17981-17993.

[4] Hoogeboom, Emiel, et al. "Autoregressive diffusion models." arXiv preprint arXiv:2110.02037 (2021).

[5] Chen, Yuzhu, et al. "Adaptive Time-Stepping Schedules for Diffusion Models." The 40th Conference on Uncertainty in Artificial Intelligence.

**Questions:**

1. As the empirical validation of the KLUB is a proxy for CDE, would it be possible to directly show that minimizing KLUB actually reduces this error?
2. Does the hierarchical breakdown strategy guarantee convergence to a globally optimal schedule?
3. The paper says “After K iterations, this hierarchical strategy yields a sampling schedule with 2^K NFEs, optimizing the schedule as the number of steps increases.” in lines 289-29. So, there is an exponential factor appearing in the number of function estimations. I wonder if it affects the overall complexity and method’s applicability, especially in resource-constrained settings.
4. I also wonder if the computational gain from this approach leads to higher memory costs. Is it this case from your observations?
5. Is there any intuition on how the method's error bounds scale with data size, or it doesn’t matter?
6. Is there any insight into the applicability of the proposed method to DDMs without fast sampling algorithms?
7. Do you think the hierarchical breakdown strategy converges to globally optimal schedules?

Small suggestions regarding the manuscript:
- There is no need to define DDMs (lines 12, 30, 94, for example) or CDE (line 156) a few times through the text if they were defined before.
- In Eq. 1 and this section, I’d suggest explicitly defining $x$ (data) and $y$ (corrupted data) variables.
- Line 133: Should it be “.” instead of “:”?
- Maybe notations in line 148 can be combined with Section 2.1 or moved there? I’d suggest the authors go through their manuscript to make sure all notations are introduced and consistent.
NFE is not defined in line 290 or earlier. Also, it is useful to explain what it means (it is the number of calls to the score NN, right?).
- Maybe it’s worth moving Section 5 to the Intro section or a separate section at the beginning of the manuscript? Also, mentioning papers [1, 2] might be helpful.
- Adding equation numbers to Algorithm 1(line 7) would be helpful.

---

> ### Author Response · Authors · 2024-11-22
>
> Thank you for acknowledging the strengths of our work, including its sound theoretical development and the fact that it requires no architectural modifications or additional inference costs.
>
> > **[W1]** The empirical analysis can be improved.
>
> We appreciate your suggestions for enhancing our empirical analysis. We addressed your points by breaking them down as follows:
>
> > **[W1-1]** Comparison with other samplers, e.g., predictor-corrector sampler.
>
> Thank you for recommending additional baselines. We conducted experiments with the predictor-corrector (PC) method, following the LDR-3 settings proposed in [1]. As shown in Figure 6, our JYS method can be applied to the PC sampler, achieving performance improvements.
>
> > **[W1-1,2, Q3, Q5]** Analysis of the computational costs and trade-off between schedule optimization time/sample quality.
>
> We appreciate your proposal for experiments that highlight the practical utility of our method. We perform an in-depth analysis of the scaling behavior with dataset size and the computational costs involved in optimization, which are discussed in detail in Global Rebuttal #2.
>
> In summary, even when the number of Monte Carlo samples is reduced to 16, there is little to no performance degradation. Moreover, optimizing with 16 samples allows the JYS sampling schedule optimization for NFE=64 to complete in approximately 45 seconds for CIFAR-10 and about 5 minutes for text generation on RTX3090 GPUs. While this is not a direct apple-to-apple comparison, we note that AYS requires about 6 GPU hours for CIFAR-10 and over 32 GPU hours for ImageNet 256×256 on RTX6000 GPUs [2]. We hope this demonstrates the efficiency and practicality of our method.
>
> > **[W1-3]** Error bars or standard deviations for the quantitative metrics.
>
> In Figure 12, we report the minimum, maximum, and mean values from experiments conducted with three random seeds on CIFAR-10 and text generation. The results show minimal differences across the three seeds, demonstrating the robustness of our metric to random seed variations. This robustness likely stems from generating a sufficiently large number of samples (10K for FID and 1024 for generative perplexity) before performing the measurements, thereby reducing variability. For CIFAR-10, the average difference between the maximum and minimum FID (10K) values is 0.37, while for text generation, the average difference in perplexity (PPL) values is 0.65—both negligibly small.
>
> > **[W1-3]** Comparison with other sampling schedule optimization methods
>
> Thank you very much for bringing this relevant reference to our attention. The paper you mentioned [3] is indeed similar to AYS, as it optimizes the sampling schedule by minimizing the KLUB proposed by [4], but it employs gradient descent (GD) with numerical differentiation for optimization, rather than the zeroth-order optimization method used in AYS. In our view, applying this idea to our study would correspond to replacing the golden-section search with GD.
>
> In fact, during our preliminary experiments, we attempted to use GD in place of the golden-section search. However, we encountered two significant challenges: first, the numerical differentiation required for GD was highly noisy; second, GD was extremely sensitive to the learning rate. The latter issue posed a considerable challenge, as we needed to determine an optimal learning rate each time we searched for a sampling step. For instance, if NFE=64, we would need to find 64 different optimal learning rates. Therefore, we chose the golden-section search, which does not involve hyperparameters that significantly affect optimization.
>
> Additionally, in response to Reviewer tVar's suggestion, we included heuristic sampling schedules that are not uniform as additional baselines (Figure 14). We found that all of these performed worse than the JYS schedule, suggesting that it may be challenging to find an optimal sampling schedule using heuristics alone.

---

> ### Author Response · Authors · 2024-11-22
>
> > **[W2]** The paper doesn't clarify how the diffusion model formulation (Eq. 1, CTMC) relates to other DDM variants from [a, b, c]. Such discussion would be appreciated.
>
> Thank you for bringing these relevant references to our attention. We have carefully reviewed them and incorporated them into our paper. Addressing them in order:
>
> - References [a] and [b] can be considered as CTMCs but are restricted to discrete-time schedules with specific values of $t$. In particular, [a] addresses special cases where the transition kernel is limited to transitions between adjacent values.
> - Reference [c] focuses on parallel transformer models, which can be understood as denoising parameterization models where tau-leaping sampling is applicable. However, the sampling method presented in [c] is challenging to represent within the CTMC framework.
>
> We are grateful for your suggestions, as they allowed us to better position our work in relation to these studies.
>
> > **[W3]** The single-state transition assumption, while mathematically convenient, limits the method's applicability
>
> Thank you for suggesting a direction to improve our research. We would like to note that in the case of an absorbing transition kernel, where only one transition is possible throughout the entire sampling process per each token, our single-state transition assumption provides an exact solution rather than an approximation. However, for other types of transition kernels, it is indeed an approximation. We acknowledge this limitation and have included this discussion in our paper, highlighting it as an important direction for future studies (Appendix A.1).
>
> > **[W5]** No code is provided for reproducibility.
>
> We assure you that we will make our code publicly available as open-source software upon publication.
>
> > **[Q1]** As the empirical validation of the KLUB is a proxy for CDE, would it be possible to directly show that minimizing KLUB actually reduces this error?
>
> Thank you for this insightful question. Unfortunately, directly measuring CDE is not practically feasible, as the KL divergence is generally intractable in high-dimensional spaces. To address this limitation, we demonstrated that our JYS method improves sampling quality across a variety of datasets and types of DDMs. These results indirectly suggest that minimizing KLUB effectively reduces CDE.
>
> > **[W4, Q2, Q6]** Does the hierarchical breakdown strategy guarantee convergence to a globally optimal schedule?
>
> Unfortunately, the hierarchical breakdown strategy is a heuristic algorithm designed to perform optimization efficiently; it is not a guaranteed algorithm for reaching the global optimum. Nevertheless, through extensive experiments, we demonstrated that the hierarchical breakdown strategy is an effective approach for reducing CDE and improving sampling quality.
>
> Additionally, as part of our preliminary experiments, we explored an alternative strategy—hierarchical merging—where smaller, fragmented sampling schedules are combined into a coarse sampling schedule. However, in our initial experiments, this approach often became stuck in local optima and failed to yield significant performance improvements.
>
> > **[Q4]** Does computational gain from technique 1, 2 leads to higher memory costs?
>
> Thank you for the opportunity to clarify the computational efficiency of our method. No, our techniques do not increase memory costs. In fact, the technique 1 and 2 reduce the number of samples required for Monte Carlo estimation, ensuring efficient performance. As mentioned earlier, our method can run smoothly on a single NVIDIA 3090 GPU. Using a GPU with more VRAM would enable a larger number of KLUB estimations to be computed simultaneously, further reducing the optimization time.
>
> > **[Q8]** Small suggestions regarding the manuscript:
>
> Thank you. We incorporated your suggestions into the revised manuscript.
>
> ---
>
> [1]: Campbell et al., A Continuous Time Framework for Discrete Denoising Models, NeurIPS22
>
> [2]: Sabour et al., Align Your Steps: Optimizing Sampling Schedules in Diffusion Models, ICML24
>
> [3]: Chen et al., Adaptive Time-Stepping Schedules for Diffusion Models, UAI24
>
> [4]: Chen et al., Sampling is as easy as learning the score: theory for diffusion models with minimal data assumptions, ICLR23
>
> [a] Santos et al., Blackout diffusion: generative diffusion models in discrete-state spaces, ICML23
>
> [b] Austin et al., Structured denoising diffusion models in discrete state-spaces, NeurIPS23
>
> [c] Hoogeboom et al., Autoregressive diffusion models, ICLR22

---

> ### Comment · Reviewer_ibjS · 2024-11-26
> **Thank you for the response**
>
> Thank you for your thoughtful response to the review. I continue to believe that this paper offers a nice method with empirical improvements over past methods and helpful insights into the limitations of past methods. The speed of the optimization is a compelling factor in assessing the practical relevance of the method. I think the insights leading to this paper may be helpful in inspiring follow-on work.

---

### Official Review · Reviewer_TF1f · 2024-11-04

**Soundness:** 2
**Presentation:** 2
**Contribution:** 2
**Rating:** 5
**Confidence:** 4

**Summary:**

This paper introduces Jump Your Steps (JYS), a novel approach that optimizes the allocation of discrete sampling timesteps. It achieves this by minimizing Compounding Decoding Error (CDE) without incurring any additional computational cost. The authors derive an upper bound on CDE and employ techniques to simplify the computation. Based on the upper bound, they propose an algorithm for searching for the optimal sampling schedule, thereby enhancing the sampling quality.

**Strengths:**

The algorithm presented in this submission is straightforward and simple. Experimental results have been provided to demonstrate the feasibility of the proposed algorithm.

**Weaknesses:**

1. The novelty of the proposed algorithm is rather limited. The authors merely transfer Align Your Steps (AYS), which is originally for image generation, to the framework of SEDD for text generation.

2. The experiments conducted on the CIFAR-10 and Countdown datasets lack practical significance. In the realm of image generation, efficient sampling methods have been well explored. For CIFAR-10, even with 256 NFEs, the proposed approach can only achieve an FID higher than 10. In contrast, some popular sampling methods for image generation, such as DPM-Solver++ (https://arxiv.org/pdf/2211.01095) and DMD (https://arxiv.org/pdf/2311.18828), can attain an FID less than 4 using only 10 or even just 1 sampling step. Additionally, the Countdown dataset is a synthetic one. I suggest that the authors place the results for CIFAR-10 and Countdown in the appendix. Moreover, as I am not familiar with music generation, it appears to me that although JYS can lead to a significant drop in Hellinger distance, the Hellinger distance for uniform sampling timesteps is already small (less than 0.4).

3. It seems to me that the practically meaningful experimental results are mainly showcased in Figure 8 for text generation. Nevertheless, upon examining Figure 8, I cannot observe a significant improvement over the use of uniform sampling timesteps.

4. Some problems in the writing:
- The right arrows used on Line 52 seem strange. Additionally, such right arrows occur multiple times in the main content. It is suggested to use normal notations instead (and I don't think there is a need to write down the whole sequence for timesteps repeatedly).
- The term "KLUB" is first seen in Figure 2 on page 4; however, its explanation is only provided in Theorem 3.1 on page 5.
- There is no necessity to write down the Sampler in Figures 6 and 7.
- In the section on "$k$-Gillespie's Algorithm" (Line 125), '$k$' apparently means updating $k$ tokens in parallel. However, in Appendix B.1 KLUB COMPUTATION, the use of '$p(t|k)$' is presented and it is explained that this is determined by the pretrained noise schedule. As a result, I am not certain whether '$t$' signifies time. If it does, then it appears inconsistent with '$k$'. Please provide a more detailed explanation about $k$-Gillespie's Algorithm.
- The condition $\mathbb{P}_{t_0} = \mathbb{Q}_{t_0}$ in the last sentence of the proof of Theorem 3.1 should be presented as an assumption in the statement of Theorem 3.1.


--------

After rebuttals:

I would like to thank the authors for their detailed responses. However, the authors have only provided Figure 13 (which has been shifted to illustrate the performance gains of JYS compared to those achieved by increasing model size) to address my major concern in W3. Currently, although I have slightly raised my score to 5, my concern about the practical significance and relevance of the proposed approach still persists. As a result, I am still inclined towards rejection.

**Questions:**

N/A

---

> ### Author Response · Authors · 2024-11-22
>
> > **[W1]** The novelty of the proposed algorithm.
>
> We are pleased to provide clarification and emphasize that our contributions go beyond a straightforward adaptation of Align Your Steps (AYS). In the global rebuttal, we outlined how Theorem 3.2 and our algorithm represent distinct and novel contributions compared to existing work.
>
> > **[W2]** CIFAR-10, Countdown, Piano experiments lack practical significance.
>
> We would like to clarify that our research does not focus on achieving state-of-the-art (SOTA) image generation performance. Instead, our goal is to diagnose sampling issues in DDMs and propose a principled, mathematically grounded approach with broad applicability. The image generation experiments are intended to validate the effectiveness of our method in improving DDM samplers, rather than optimizing FID scores to SOTA levels. In this context, our experiments on CIFAR-10, Countdown, and Piano demonstrate substantial performance improvements, showing that the JYS schedule achieves comparable performance to the uniform schedule while requiring only about half the NFE.
>
> Additionally, since JYS is an algorithm that enhances sampling, achieving SOTA performance largely depends on the pre-trained models used. JYS is a generic, plug-and-play method that is agnostic to specific models and generally improves performance. This means that if we have a well-trained model, JYS can provide an even better generation performance.
>
> > **[W3]** In text generation, JYS do not lead to a significant improvement over the use of uniform sampling timesteps.
>
> We respectfully disagree with the assertion that JYS does not offer significant improvements in text generation. To demonstrate the impact of JYS, we compared its performance gains to those achieved by increasing model size. In Figure 13, we show the relative reduction in generative perplexity (ppl) when transitioning from a uniform to a JYS sampling schedule at each NFE, compared to the reduction observed when scaling the model from SEDD-small to SEDD-medium. The results indicate that at NFE=16, the performance improvement from JYS is nearly equivalent to that of increasing the model size, and even at NFE=256, JYS achieves a 15% improvement compared to increasing the model size.
>
> Given that our method requires neither fine-tuning nor additional inference costs, we believe these performance gains are highly significant. Moreover, we note that at NFE=16, the SEDD-small model achieves a perplexity (ppl) of 130, which is substantially lower than the reported ppl of 240 for GPT-2 small, further underscoring its practical significance.
>
> > **[W4]** Some suggestions to improve writing.
>
> Thank you for your detailed suggestions. We have addressed each of your points below:
>
> - Notation: Sampling schedule
>
> We sincerely appreciate your feedback regarding our notation. To address any potential confusion, we have added a footnote clarifying that the right arrow notation denotes the sampling step. If you have any suggestions for a notation that might be clearer than the right arrow ($s \rightarrow t$) to represent sampling from timestep $s$ to $t$, we would be most grateful to consider them in our revision. We chose the sampling schedule notation as $\{T, \rightarrow t_1 \rightarrow \cdots \rightarrow 0\}$ because it naturally extends single sampling step to the entire set of timesteps, providing consistency throughout the paper.
>
> - The term "KLUB" is first seen in Figure 2 on page 4.
>
> Thank you for pointing this out. We updated the caption of Figure 2 to clarify that KLUB is defined in Theorem 3.1, ensuring readers are directed to the correct definition.
>
> - Please provide a more detailed explanation about k-Gillespie's Algorithm.
>
> First of all, we have expanded the description of the k-Gillespie algorithm in the appendix to enhance clarity (Appendix C.3). To answer your question, in $p(t|k)$, $k$ represents the number of unmasked tokens, and $t$ is the timestep. The noise schedule determines the probability that each token is perturbed to masked token at given timestep. By using noise schedule, we can compute the probability that $k$ tokens are perturbed at time $t$, i.e., $p(k|t)$. By reversing the process of finding $k$, we can use it to determine $t$, allowing us to sample $t$ from $p(t|k)$. We will provide a more detailed explanation of $p(t|k)$ in Appendix C.1.
>
> - $\mathbb{P}_{t_0} = \mathbb{Q}_{t_0}$ should be presented as an assumption in the statement of Theorem 3.1.
>
> Thank you for highlighting this potential confusion. In the proof of Theorem 3.1, $t_0$ is the starting timestep for sampling, which corresponds to $T$. Since $\mathbb{P}_T = \mathbb{Q}_T = \pi$ in DDMs, where $\pi$ is the terminal distribution of forward CTMC, this condition is always satisfied and does not require an additional assumption. To clarify, we have made this explicit in the proof in the revised manuscript.

---

> ### Author Response · Authors · 2024-11-25
>
> Thank you for your response! I’m glad to hear that Figure 13 partially addresses your concern regarding the practical significance of our method. If you have any remaining concerns about the relevance of our proposed approach, I would greatly appreciate it if you could elaborate further so we can have the opportunity to address them.
>
> Once again, thank you for your time and efforts.

---

> > ### Comment · Reviewer_TF1f · 2024-11-25
> >
> > I want to clarify that I raised the score because of the responses to other weaknesses. Figure 13 does not address my concern regarding the practical significance, and my concern about the practical significance and relevance of the proposed approach still persists.

---

> > > ### Author Response · Authors · 2024-12-01
> > >
> > > Dear Reviewer TF1f,
> > >
> > > Thank you for your valuable feedback.
> > >
> > > Due to time and resource constraints, we were limited in our ability to verify our methods on a broader range of discrete diffusion models, as there are relatively few open-sourced state-of-the-art implementations with well-structured code available. We appreciate your concern about practical significance, and we aim to conduct additional experiments on ImageNet before the publication to address this limitation.
> > >
> > > Thank you again for your valuable feedback to our work.

---

### Official Review · Reviewer_tVar · 2024-11-04

**Soundness:** 2
**Presentation:** 3
**Contribution:** 2
**Rating:** 5
**Confidence:** 4

**Summary:**

This paper proposes a method, Jump Your Steps, to optimize the discretization schedule of discrete diffusion models, by minimizing the compounding decoding error (CDE). The authors provide a KL-divergence upper bound (KLUB) for different continuous-time Markov chains of discrete diffusion models and propose techniques to efficiently approximate the timesteps minimizing KLUB.

**Strengths:**

This paper proposes an approach to improve the sampling quality and efficiency by optimizing the sampling schedule. This paper is well-written and easy to follow. The sampling and acceleration of diffusion models is an interesting topic for the community.

**Weaknesses:**

- Theorem 3.1 seems to be standard entropy bounds, and Theorem 3.2 seems to directly follow from Equations 3.2 and 3.4 of Ding & Ning (2021) and Theorem 3.2 of Sabour et al. (2024). The authors should illustrate their technical contributions and the novelty of their theoretical results with a comparison to the previous literature.
- The authors only compare the JYS schedule with the uniform schedule. Additional experiments on other schedules, e.g. EDM, Linear LogSNR, and Cosine LogSNR, should be presented.

**Questions:**

- The proposed scheduling strategy only considers $2^K$ numbers of function evaluations (NFE). How can the schedule extend to other values of NFE?
- In Figure 5, is there any intuition to explain the error increase of the $k$-Gillespie sampler with the JYS schedule when NFE=16?
- In Figure 10, the comparison between uniform and JYS is not significant for NFE=32, 128, or 256. Please consider providing additional experiments on other image datasets, e.g., toy 2D datasets in Sabour et al. (2024).

---

> ### Author Response · Authors · 2024-11-22
>
> Thank you for acknowledging that our paper is well-written and easy to follow, and also handle interesting topic, which is acceleration of diffusion models.
>
> > **[W1]** The technical contributions and the novelty of the theoretical results (Theorem 3.1 and 3.2) in comparison to the existing literature.
>
> Thank you for giving us the opportunity to clarify our contributions. Please refer to the Global rebuttal #1 for a detailed discussion on the novelty of Theorem 3.2 and our algorithms. Here, we will focus on articulating the contribution of Theorem 3.1.
>
> In Theorem 3.1, we establish a crucial link between Compounding Decoding Error (CDE) and sampling error ($D_{KL}(\mathbb{P}_0||\mathbb{Q}_0)$). While CDE was introduced in [1] and several studies have attempted to address it, **no prior work has formally defined CDE or explained its impact on sample quality, leaving a significant gap** in understanding. Our work fills this gap by rigorously demonstrating that CDE is meaningfully related to sampling quality. While our proof leverages the well-known inequality of KL divergence, the novelty of our theorem lies in uncovering a previously unexplored insight: the relationship between CDE and sample quality.
>
> > **[W2]** Additional baseline results on other schedules, such as EDM, Linear LogSNR, and Cosine LogSNR, beyond the uniform schedule.
>
> Thank you for suggesting additional experiments to make our empirical results more compelling. First, we would like to kindly clarify that our research specifically focuses on *sampling schedules*. The methods you mentioned, such as EDM, Linear LogSNR, and Cosine LogSNR, are *noise schedules*, which are related but have a different focus.
>
> Nevertheless, to provide a more comprehensive evaluation, we conducted additional experiments comparing our approach with more heuristic sampling schedules beyond the uniform baseline. Specifically, we experimented with a sampling schedule that has the same SNR schedule as Cosine LogSNR, as well as heuristic schedules inspired from timesteps sampling in [2] that allocate more sampling steps in the middle (Sigmoid) or both end (Logit). As shown in Figure 14, the JYS schedule consistently achieves the best performance across all NFEs among various heuristic sampling schedules. This result highlights the difficulty of improving sampling schedules using only heuristic methods.
>
> > **[Q1]** The possibility of JYS sampling schedule other than $\text{NFE}= 2^K$
>
> Thank you for suggesting the extension of our method to other scenarios. Unfortunately, due to the hierarchical breakdown strategy, which doubles the steps at each level, constructing a JYS sampling schedule for $\text{NFE}$ values other than $2^K$ is currently challenging. We included this discussion in the limitations section and hope that future research will address this constraint.
>
> > **[Q2]** Explanation of the error increase of the k-Gillespie sampler with the JYS schedule when NFE=16 (Figure 5).
>
> Thank you for your detailed question. We believe the observed error increase with the k-Gillespie sampler using the JYS schedule at $\text{NFE} = 16$ was caused by issues in the model's output at specific timesteps during training. After retraining the model, we confirmed that the issue was resolved (Figure 5).
>
> > **[Q3]** Additional experiments on other image datasets, e.g., toy 2D datasets in Sabour et al. (2024).
>
> Thank you for suggesting a more intuitive way to present our results. **Figures 15–17 showcase qualitative comparisons on a toy 2D dataset** to illustrate the performance of the JYS and uniform schedules. The results clearly show that, across all toy dataset types and NFE values, the JYS schedule consistently produces samples closer to the ground truth distribution than the uniform schedule.
>
> [1] Lezama et al., Discrete Predictor-Corrector Diffusion Models for Image Synthesis, ICLR23
> [2]: Esser et al., Scaling Rectified Flow Transformers for High-Resolution Image Synthesis, ICML23

---

> > ### Comment · Reviewer_tVar · 2024-11-25
> >
> > Thank you for your response, especially the comprehensive presentation of toy experiments in Figures 15-17, which clarifies most of my questions.
> >
> > My remaining concern is that the improvement by JYS may not be very significant. For instance, in Figures 14 and 15-17 it seems that by using twice NFE, the uniform schedule could have a similar performance to JYS. It indicates that JYS can only reduce NFE by half, in order to achieve similar performance to the uniform schedule. Combining this with the fact that JYS can only apply NFE=$2^k$ (while other schedules may apply any NFE) may limit the practical improvement of JYS, e.g., other schedules with NFE=$2^k+\delta$ (for some small value of $\delta$) may achieve similar performance to JYS with NFE=$2^k$.

---

> ### Author Response · Authors · 2024-11-26
>
> Thank you for waiting for our response. We also appreciate you bringing up a practical scenario that we had overlooked.
>
> To address your concern, we proposed a method to apply the JYS schedule even in situations where NFE$ \ne 2^K$. This method shows that even when NFE$ \ne 2^K$, **JYS schedule can achieve performance comparable to uniform sampling with only half the NFE**.
>
> ---
>
> First of all, as you mentioned, in the case where NFE$ = 2^K$, it seems that JYS can achieve performance comparable to the uniform method with only half the NFE. However, previously, since the JYS schedule exists only for NFE$ = 2^K$, problems can arise. For example, in an extreme case where one must achieve the performance of the uniform method with NFE$ = 2^K + 1$, one would have to use JYS with NFE$ = 2^K$, which is highly inefficient. In other words, flexibility seems important in practical applications.
>
> Therefore, we proposed a method to apply the JYS schedule even in situations where NFE$ \ne 2^K$. It is a simple method: we use only a subset of the already optimized JYS schedule. The method is straightforward. For example, suppose we want to consider NFE$ = 48$. In this case, we first use all the timesteps corresponding to NFE$ = 32$, and then include additional timesteps $t_{2i}$ from the remaining ones in the order of decreasing $\text{KLUB } \left( Q^{t_{2i-1} \rightarrow t_{2i}\rightarrow t_{2i+1}} \,\|\, Q^{t_{2i-1}\rightarrow t_{2i+1}} \right)$ values. This can be understood as selecting $t$ from those that reduce the CDE as much as possible, which was our origin motivation.
>
> The results can be found in **Figure 15** of the revised PDF. (Note that we have uploaded a new figure, so it is different from the previous Figure 15.) Notably, even when NFE is not a power of two, the JYS sampling schedule achieves comparable performance to the uniform baseline with only half the NFE. We included precise method and results at Appendix E.5.
>
> Once again, thank you for giving us the opportunity to improve our method.

---

> ### Author Response · Authors · 2024-12-01
>
> Dear Reviewer tVar,
>
> We wanted to kindly follow up regarding your previous concern about the JYS sampling schedule where NFE ≠ 2^K. We have conducted additional experiments addressing this point, and we are pleased to share that the results can be found in the Revised PDF Figure 15. The findings demonstrate that we can achieve comparable performance with JYS while using only approximately half the NFE compared to the Uniform, even NFE ≠ 2^K.
>
> We would be very grateful if you could let me know if you have any remaining concerns or questions. We are committed to ensuring all aspects of our work meet your expectations.
>
> Thank you again for your valuable feedback to our work.

---

> ### Comment · Reviewer_tVar · 2024-12-02
>
> Thank you for your response. I think the quantitative evaluations and comparisons in the current version are comprehensive. However, as noted in my initial review Q3 and also raised by reviewer TF1f, I remain concerned about the practical significance of the JYS, especially in Figure 19, where the comparisons between uniform and JYS are not significant for NFE=32, 128, or 256. More precisely, when uniform scheduling starts to be able to generate meaningful samples (i.e., NFE=128), JYS with half NFE=64 cannot generate meaningful samples.

---

> ### Author Response · Authors · 2024-12-02
>
> Thank you for your thoughtful feedback and detailed review of our work.
>
> We appreciate the opportunity to clarify an important aspect of our research. Our study focuses on JYS's ability to enhance performance while given the **same NFE**, rather than achieving equivalent performance at half the NFE. We believe this distinction is crucial for accurately evaluating the method's contributions.
>
> Regarding the visual assessment, while we appreciate your careful examination of the samples in Figure 19, we believe it's worth considering both qualitative and quantitative measures. The FID improvements shown in Figure 6, particularly at NFE = 128 and 256, demonstrate meaningful progress according to this widely-accepted metric. Could you elaborate on why you find the FID improvements unconvincing? We'd also like to understand any concerns you have about using FID to measure image generation quality.
>
> We sincerely appreciate your valuable feedback and the opportunity to engage in this discussion.

---

> > ### Comment · Reviewer_tVar · 2024-12-02
> >
> > - I understand the primary goal is to compare JYS and uniform with the same NFE. However, to evaluate the effectiveness of JYS, I believe it is also important to study what is the minimal value of NFE that JYS needs to achieve similar performance as uniform, as you stated in your revision that "the JYS sampling schedule achieves comparable performance to the uniform baseline with only half the NFE" (line 1369-1371), and also in your response to Review TF1f  that "In this context, our experiments on CIFAR-10, Countdown, and Piano demonstrate substantial performance improvements, showing that the JYS schedule achieves comparable performance to the uniform schedule while requiring only about half the NFE."
> >
> > - I understand that FID is the standard approach to quantitatively evaluate the performance of generative models. However, I also believe that FID *sometimes* cannot fully reflect the quality as seen by humans. In Figure 19, the quality of JYS with NFE=64 is clearly worse (at least for me, to be elaborated in the next point) than uniform with NFE=128, which seems to be inconsistent with your claim in line 1369-1371 and your response to Review TF1f.
> >
> > - Regarding Figure 19, my identification of these images is:
> >
> > Jump your steps (NFE=64):
> >
> > (vague), (vague), (vague), bird, (blurred ship), (blurred airplane), ship, (vague)
> >
> > (vague), (vague), (vague), (vague), (blurred deer), (vague), (vague), (blurred frog?)
> >
> > Uniform (NFE=128):
> >
> > Truck, ship, dog, automobile, automobile, ship, (vague), (vague)
> >
> > Airplane, ship, (vague), (blurred automobile), (blurred cat?), (vague), (vague), ship
> >
> > Considering the number of images I can identify, my identification suggests that in Figure 19, the quality of JYS (NFE=64) is worse than uniform (NFE=128). Therefore, as Figures 6 (which only considers NFE=64) and 19 are the only visual results in this work, I am not fully convinced of the practical significance of JYS.

---

> ### Author Response · Authors · 2024-12-02
>
> Thank you for your valuable feedback.
>
> > I believe it is also important to study what is the minimal value of NFE that JYS needs to achieve similar performance as uniform
>
> First, we would like to clarify my statement about JYS achieving similar performance with roughly half the NFE, as it may have been somewhat of an overclaim. Through precise FID value comparisons, we found that JYS with NFE values of 40, 96, and 112 achieves equivalent or better performance compared to Uniform NFE of 64, 128, and 256 respectively. While this shows that we can reduce NFE by more than half for NFE = 256, the reduction for NFE = 64 and 128 is slightly less than half.
>
> To address your concern, we quickly generated qualitative figure comparisons between Uniform NFE = 128, 256 and JYS NFE = 96, 112, which visually confirmed no perceptible difference in performance. Now, you can check the figure at the anonymous link in the comment below.
>
> We had assumed that comparing performance using either equal NFE values (Figure 15, fix x-axis) or equal performance levels (Figure 15, fix y-axis) would convey the same message, but we overlooked that some readers might interpret this differently. We appreciate your suggestion as it helps make our paper more solid, and we are grateful for your contribution to improving our manuscript.

---

> > ### Author Response · Authors · 2024-12-03
> >
> > Dear Reviewer tVar,
> >
> > Thank you for your patience. Using the ICLR review policy that allows sharing results through anonymized links, we would like to share the comparison results between CIFAR-10 Uniform NFE = 128, 256 and JYS NFE = 96, 112.
> >
> > https://docs.google.com/document/d/e/2PACX-1vSGVkbXPXalHpb5OxfzY4qp3mxbGKrVum8XLpiZrQb8zPKi1w5k9F00_wKY2N47DMAhkibEETsCOjXp/pub
> >
> > This qualitative figure demonstrates that we can achieve the comparable performance on CIFAR-10 with lower NFE. We hope this result helps address your consideration.

---

> > > ### Author Response · Authors · 2024-12-04
> > > **Still looking forward to the reviewer's feedback**
> > >
> > > Dear Reviewer tVar,
> > >
> > > As the deadline for reviewers to post a message has recently passed, we understand that it is no longer possible to provide an official comment. **However, we would be deeply grateful if the reviewer could kindly edit the original review to let us know whether our rebuttal and the revised paper adequately address the questions and concerns previously raised**. Your feedback would be invaluable in helping us further improve our submission.

---

### Author Response · Authors · 2024-11-22

We thank each reviewer for their careful reading of our work and their thoughtful feedback.

We appreciate the positive remarks, including that our paper is well-written and easy to follow (tVar), straightforward and simple (TF1f), and novel (ibjS, VTwh) with sound theoretical developments (ibjS). We are also grateful for the recognition of our approach, which introduces performance improvment without no architectural changes or added inference costs (ibjS).

We have updated the manuscript following the suggestions in the reviews and marked the additions in blue to facilitate finding these passages. All figures mentioned in the global and individual reviewer rebuttals refer to those in the revised PDF.

---

### 1. Difference from align your steps

> **Reviewer tVar and TF1f asked us to clarify the distinct contribution of our work compared to Align Your Steps (AYS) [1].**

We are pleased to provide clarification and emphasize that our contributions extend beyond a simple adaptation of AYS.

First, our paper presents an original theoretical contribution. In Theorem 3.2, we derive an upper bound for the sampling error of discrete diffusion models (DDMs), referred to as KLUB, using Girsanov's theorem for continuous-time Markov chains (CTMC) from Ding & Ning (2021) [2]. To the best of our knowledge, this is the first study to characterize the sampling error of DDMs using KL divergence. Notably, our result offers a key insight: the sampling error of DDMs can be elegantly expressed as **a ratio of the rate matrix**, introducing a novel framework for error analysis.

We respectfully point out that results from continuous diffusion models (CDMs) cannot be directly transferred to DDMs because they are based on fundamentally different mathematical frameworks. Specifically, the KLUB for stochastic differential equations (SDEs) does not directly apply to continuous-time Markov chains (CTMCs). In SDEs, the KLUB derived by Chen et al. (2022) [3] depends on the difference between the drift terms of two SDEs (i.e., $||f_1 - f_2||_2^2$). In contrast, for CTMCs, the KLUB is defined by the ratio of the rate matrices between two CTMCs (i.e., $\log (R_1 / R_2)$), which cannot be obtained through simple adaptation. While AYS [2] demonstrates how KLUB can be utilized for optimizing sampling schedules, our work shares only the general idea of using KLUB for sampling schedule optimization; the theoretical development and application are entirely different.

Second, we provide a distinct technical contribution that sets our work apart from AYS. In CDMs, KLUB can be directly computed numerically, whereas directly calculating KLUB for DDMs is infeasible with practical computational resources. To address this challenge, we developed hierarchical breakdown strategy (Section 3.4) with Techniques 1 and 2 (Section 3.5), leading to highly efficient optimization algorithms (Algorithms 1 and 2). This advancement not only makes sampling schedule optimization practical for DDMs but also significantly reduces computational costs. For example, for CIFAR-10, while AYS requires approximately 6 GPU hours to optimize the sampling schedule for NFE=50, **our approach completes the optimization in under 45 seconds for NFE=64**. We discuss our algorithm's efficiency in greater detail in Global Rebuttal #2.

Lastly, we would like to emphasize our focus on addressing the unique challenge of accelerating DDMs. While CDMs benefit from established numerical solvers like DPM and DPM++, which greatly improve sampling efficiency and image quality, such solutions are not available for discrete data. In discrete settings, there are no equivalent particle-wise differential equation solvers suitable for DDM sampling. Therefore, we analyzed the specific problem of Compounding Decoding Error (CDE) inherent in DDM sampling and proposed a principled, mathematically grounded solution. Consequently, we developed an alternative approach: optimizing the sampling schedule to mitigate CDE. We have tested our method across various modalities and transition kernels, demonstrating its broad applicability and effectiveness.

[1]: Sabour et al., Align Your Steps: Optimizing Sampling Schedules in Diffusion Models, ICML24
[2]: Ding & Ning, Markov chain approximation and measure change for time- inhomogeneous stochastic processes, Applied Mathematics and Computation, 2021
[3]: Chen et al., Sampling is as easy as learning the score: theory for diffusion models with minimal data assumptions, ICLR23

---

> ### Author Response · Authors · 2024-11-22
>
> ### 2. Efficiency
>
> > **Reviewer ibjS asked us to study the trade-off between schedule optimization time and sample quality gains, and overall efficiency of the algorithm 1.**
>
> Thank you for proposing an insightful experiment that highlights the practical utility of our algorithm. When optimizing the sampling schedule, there is indeed a trade-off: increasing the number of Monte Carlo samples for KLUB estimation improves reliability but also increases computational demands. Thus, our need to use the minimum number of samples necessary while maintaining reliable performance.
>
> We conducted experiments on CIFAR-10 and a text generation task to evaluate this trade-off (see Figure 10). Our findings show that reducing the number of Monte Carlo samples to as few as 16 does not result in a significant performance drop. We hypothesize that the sampling schedule optimization is robust with fewer samples because the KLUB, which our optimization aims to maximize, does not vary greatly across sample trajectories (Figure 4).
>
> Based on these results, we measured the time required for the JYS sampling schedule optimization on a practical setup: a single 24GB NVIDIA RTX 3090 GPU (Figure 11). Note that with more GPUs, the Monte Carlo sampling could be parallelized, further reducing time. For sampling schedule NFE=64, **the optimization took only 45 seconds on CIFAR-10, and 5 minutes on text generation model (SEDD-small).** In contrast, AYS required approximately 6 GPU hours for NFE=50 on CIFAR-10 and 32 GPU hours for ImageNet 256x256 with RTX6000.
>
> Importantly, this time cost applies only to the initial sampling schedule optimization; once optimized, there is no additional inference cost.

---

### Author Response · Authors · 2024-11-25

We sincerely thank the reviewers and area chair for their efforts and would like to request any missing responses to our initial rebuttal.

Given the extensive experiments in our rebuttal, we understand that reviewing the revised manuscript may be challenging. To assist, we are providing a brief summary of the newly added appendix sections:

- **Appendix A.1**: Limitations of JYS (NFE = 2^K, single-state assumption)
- **Appendix A.2**: Discussion on JYS design choices (Ignoring Accumulated Error, Golden Section Search, Hierarchical Breakdown)
- **Appendix B.1 & B.2**: Equality conditions of Theorems 3.1 and 3.2, Tightness comparison with ELBO
- **Appendix B.4**: Derivation for Technique 1 approximation
- **Appendix C.1**: JYS optimization for k-Gillespie
- **Appendix C.3**: Comprehensive explanation and algorithm for k-Gillespie
- **Appendix E.1**: Analysis of JYS computation cost
- **Appendix E.2**: Error bars and standard deviation for evaluation metrics
- **Appendix E.3**: Comparison of JYS performance improvement with model size scaling
- **Appendix E.4**: Comparison with other heuristic sampling schedules
- **Appendix E.5**: Qualitative results on Toy 2D dataset

We are grateful for your continued support and are happy to provide any further clarifications if needed.

---

### Meta-Review · Area_Chair_Mrwb · 2024-12-22

**Metareview:**

This work addresses the challenge of slow sampling speeds in discrete diffusion models (DDMs) by introducing Jump Your Steps (JYS), a novel method to optimize discrete sampling timesteps. JYS minimizes Compounding Decoding Error (CDE), a key issue in parallel sampling methods like τ-leaping, which degrade sample quality. The authors derive a practical upper bound on CDE and propose an efficient algorithm to determine an optimal sampling schedule, all without additional computational cost. Experiments across image, music, and text generation demonstrate that JYS significantly improves sampling quality. Following the author response and author-reviewer discussions, the reviews are still mixed. Considering the substantial theoretical contributions, which outweigh the relatively weaker empirical experiments, I recommend to accept.

In the final version, I encourage the authors to include a discussion on DNDM, proposed by [Chen et al., 2024]. This closely related work shares a similar concept of utilizing predetermined transition times (i.e., allocating discrete sampling timesteps) to accelerate the sampling process in discrete diffusion models.

[Chen et al. 2024] Fast Sampling via Discrete Non-Markov Diffusion Models with Predetermined Transition Time

**Additional Comments On Reviewer Discussion:**

Following the author response and subsequent discussions, the reviews remain mixed. Reviewer TF1f stated, “Currently, although I have slightly raised my score to 5, my concern about the practical significance and relevance of the proposed approach still persists. As a result, I am still inclined towards rejection.” Similarly, Reviewer tVar expressed concerns regarding the experiments. However, the other two reviewers strongly supported the submission. Considering the substantial theoretical contributions, which outweigh the relatively weaker empirical experiments, I recommend to accept.

---

### Decision · Program_Chairs · 2025-01-22

Accept (Poster)